# Temporal dynamic reorganization of 3D chromatin architecture in hormone-induced breast cancer and endocrine resistance

Yufan Zhou[1], Diana L. Gerrard[2], Junbai Wang [3], Tian Li[1], Yini Yang[1], Andrew J. Fritz[4], Mahitha Rajendran[5,6,7,8], Xiaoyong Fu[5,6,7,8], Rachel Schiff[5,6,7,8], Shili Lin[9], Seth Frietze [2] & Victor X. Jin[1]

Recent studies have demonstrated that chromatin architecture is linked to the progression of cancers. However, the roles of 3D structure and its dynamics in hormone-dependent breast cancer and endocrine resistance are largely unknown. Here we report the dynamics of 3D chromatin structure across a time course of estradiol (E2) stimulation in human estrogen receptor α (ERα)-positive breast cancer cells. We identified subsets of temporally highly dynamic compartments predominantly associated with active open chromatin and found that these highly dynamic compartments showed higher alteration in tamoxifen-resistant breast cancer cells. Remarkably, these compartments are characterized by active chromatin states, and enhanced ERα binding but decreased transcription factor CCCTC-binding factor (CTCF) binding. We finally identified a set of ERα-bound promoter–enhancer looping genes enclosed within altered domains that are enriched with cancer invasion, aggressiveness or metabolism signaling pathways. This large-scale analysis expands our understanding of high-order temporal chromatin reorganization underlying hormone-dependent breast cancer.

[1] Department of Molecular Medicine, University of Texas Health San Antonio, San Antonio, TX 78229, USA. [2] MLRS Department, University of Vermont, Burlington, VT 05405, USA. [3] Department of Pathology, Oslo University Hospital—Norwegian Radium Hospital, 0310 Montebello, Oslo, Norway. [4] Department of Biochemistry, University of Vermont, Burlington, VT 05405, USA. [5] Department of Molecular and Cellular Biology, Baylor College of Medicine, Houston, TX 77030, USA. [6] Department of Medicine, Baylor College of Medicine, Houston, TX 77030, USA. [7] Lester and Sue Smith Breast Center, Baylor College of Medicine, Houston, TX 77030, USA. [8] Dan L. Duncan Comprehensive Cancer Center, Baylor College of Medicine, Houston, TX 77030, USA. [9] Department of Statistics, The Ohio State University, Columbus, OH 43210, USA. These authors contributed equally: Yufan Zhou, Diana L. Gerrard. Correspondence and requests for materials should be addressed to S.F. (email: seth.frietze@med.uvm.edu) or to V.X.J. (email: jinv@uthscsa.edu)

Numerous efforts have been devoted to reveal the basic principle of three dimensional (3D) chromatin architecture and genome organization inside the cell nucleus among various mammalian genomes[1–8]. One prominent structural feature of the genome organization is the formation of various types of chromosomal domains[9] defined as spatial compartments[1,10], topologically association domains (TAD)[3] or lamina-associated domains (LAD)[11]. The discrete TADs ranging from several hundreds of kilobases (Kb) to several megabases (Mb) are usually stable in diverse cell types and highly conserved across different mammalian species, suggesting that they are inherent and important function units of mammalian genomes[12,13]. By contrast, spatial compartments comprised of two types, compartment A or B, form an alternating pattern of active and inactive domains along chromosome. Their sizes usually range around 5 Mb size characterized by genomic features associated with transcriptional activity, such as chromatin accessibility, active or repressive histone marks, gene density, GC content and repetitive regions[14,15]. Furthermore, A and B compartments show tissue- or cell-type specific that are correlated with cell-type specific gene expression patterns[16,17]. However, a recent study finds A or B compartments may be much smaller in size at a couple of hundred Kb by using improved Hi-C protocols in higher resolution maps[18], which are similar in size to the topologically constrained domains[19]. It is also increasingly recognized spatial compartments and TADs are fundamentally two independent chromosomal organization modes[20,21], thus disputing the common notion of a hierarchical folding principle that TADs are the building blocks of larger compartment domains.

Recent efforts have focused on understanding the relationship between higher-order structures and human development and diseases[22–24]. For instance, new studies demonstrated that the reprogramming of high-order structures of both the paternal and maternal genomes gradually occurs during early mammalian development[23,24]. Another study showed that disorganization of prostate cancer 3D genome architecture occurs coincident with long-range epigenetically activated or silenced regions of concordant gene transcription[25]. Despite the advances in our knowledge of 3D genome regulation, several critical questions remain to be answered in the field. For example, how stable or dynamic are chromosome domains upon signaling stimuli as cells respond to external cues? To what extent do these changes affect establishing or re-establishing the compartmentalized architecture? What degree of impact do the master or key transcription factors in a particular cell system have on chromatin reorganization? What are the roles of chromatin architecture in governing the progression of human diseases, such as cancers?

Estrogen (E2) signaling plays a crucial role in driving estrogen receptor α-positive (ERα+) breast cancer cell growth and proliferation[26,27]. The cellular response to E2 induction is characterized by timed and coordinated transcriptional regulation primarily mediated by ERα. Thus, it has been frequently used as a model system to illustrate the mechanisms underlying transcriptional controls in cancer development and progression as well as in fundamental biological process[28–32]. Using genome-wide approaches, we and others demonstrated there were very little overlaps of ERα targeted genes in breast cancer cells versus acquired endocrine-resistant breast cancer cells indicating distinct transcriptional regulatory mechanisms underlying endocrine resistance[33–37]. In a recent study, we used a 3C-based high throughput protocol to identify two densely mapped distant estrogen response element (DERE) regions which were frequently amplified in ERα+ breast cancer[38,39]. Interestingly, these aberrantly amplified DEREs deregulated target gene expression linked to cancer development and tamoxifen resistance. However, the roles of 3D structure and its dynamics in hormone-dependent breast cancer and endocrine resistance are largely unknown.

To establish a basis for data-driven learning and modeling of the temporal dynamics and 3D chromatin reorganization, we applied tethered chromatin conformation (TCC), a modified Hi-C protocol[40] for high depth sequencing. We performed TCC in a time-series of E2-induction in the ERα+ breast cancer cell line, MCF7 as well as the tamoxifen-resistant MCF7 (TamR) cell line. Here, we present a time-series of genome-wide maps of chromatin contacts, identify the temporal dynamic patterns of chromatin compartments, compare the patterns between MCF7 and TamR cells and examine the enrichment of ERα and CTCF binding as well as five active and repressive histone marks in the patterns. We further identify ERα-bound promoter–enhancer (ERα-PE) looping genes enclosed within TamR altered dynamic compartments. These large 3D-scale chromatin data provide a rich resource for studying the basic characteristics of hormone-dependent breast cancers and provide further insight into the mechanisms of tamoxifen resistance.

## Results

### Re-compartmentalization of chromatin at early E2 treatment.

Despite many studies demonstrating that E2 induces the highest levels of ERα binding and gene activity around 45 min to 1 h[28,30], little has been done to comprehensively characterize the changes of chromatin architecture in MCF7 cells at a genome-wide manner. In this study, we conducted TCC analysis in hormone-starved MCF7 cells (T0) and compared this to MCF7 cells treated with 1 h of E2 (T1) (Supplementary Table 1). The Pearson correlation of chromatin interactions showed that biological replicates are largely correlated for each treatment at different resolutions (Fig. 1a, Supplementary Fig. 1), illustrating a good quality of TCC data. We thus combined two replicates at either time point to identify chromosome compartments with HiCLib[10] (Fig. 1b, Supplementary Figs. 2–5). With a sequencing depth around 200 million (75 million uniquely mapped) paired-end reads for each data set, we expect the resolution of compartments around 40–50 Kb[41]. Surprisingly, the genomic size of a majority of compartments is either smaller than 1 Mb or between 1 and 2 Mb, and very few is larger than 5 Mb (Fig. 1c, d). Our data seems in stark contradiction to the compartment size of larger than 2 Mb found at earlier studies[1,10], but is consistent with newly reported studies when using improved Hi-C protocols in higher sequencing depth[18]. We found a similar number of compartments, 2067 and 2039, in the untreated and E2-treated cells respectively, where approximately half consisted of compartment A (active chromosome domains) and half were compartment B (inactive chromosome domains) at each time point (Fig. 1e). We then compared the compartments between untreated and E2-treated cells, and found that the type of compartments drastically changes following E2 treatment. The number of common or conserved compartments between treatments increase from 28%, 55%, 74% to 78% using different bin sizes of 100 Kb, 250 Kb, 500 Kb and 1 Mb, respectively (Fig. 1f). Our results at lower resolution (500 Kb or 1 Mb) are in line with many other studies that claimed ~80% conserved domains among different conditions or cell types[3,12]. However, at a higher resolution of 100 Kb, we identified 576 Common compartments between untreated and E2-treated cells and 1463 Transit compartments sensitive to E2 treatment and shifted in size or flipped between A/B compartments. Of Transit compartments, 743 (51%) of them shift only 100 Kb and 247 (17%) shift more than 400 Kb while 116 (8%) flipped between A/B compartments (Fig. 1g). These data demonstrate a re-compartmentalization of higher-order chromatin domains following 1 h of E2 treatment.

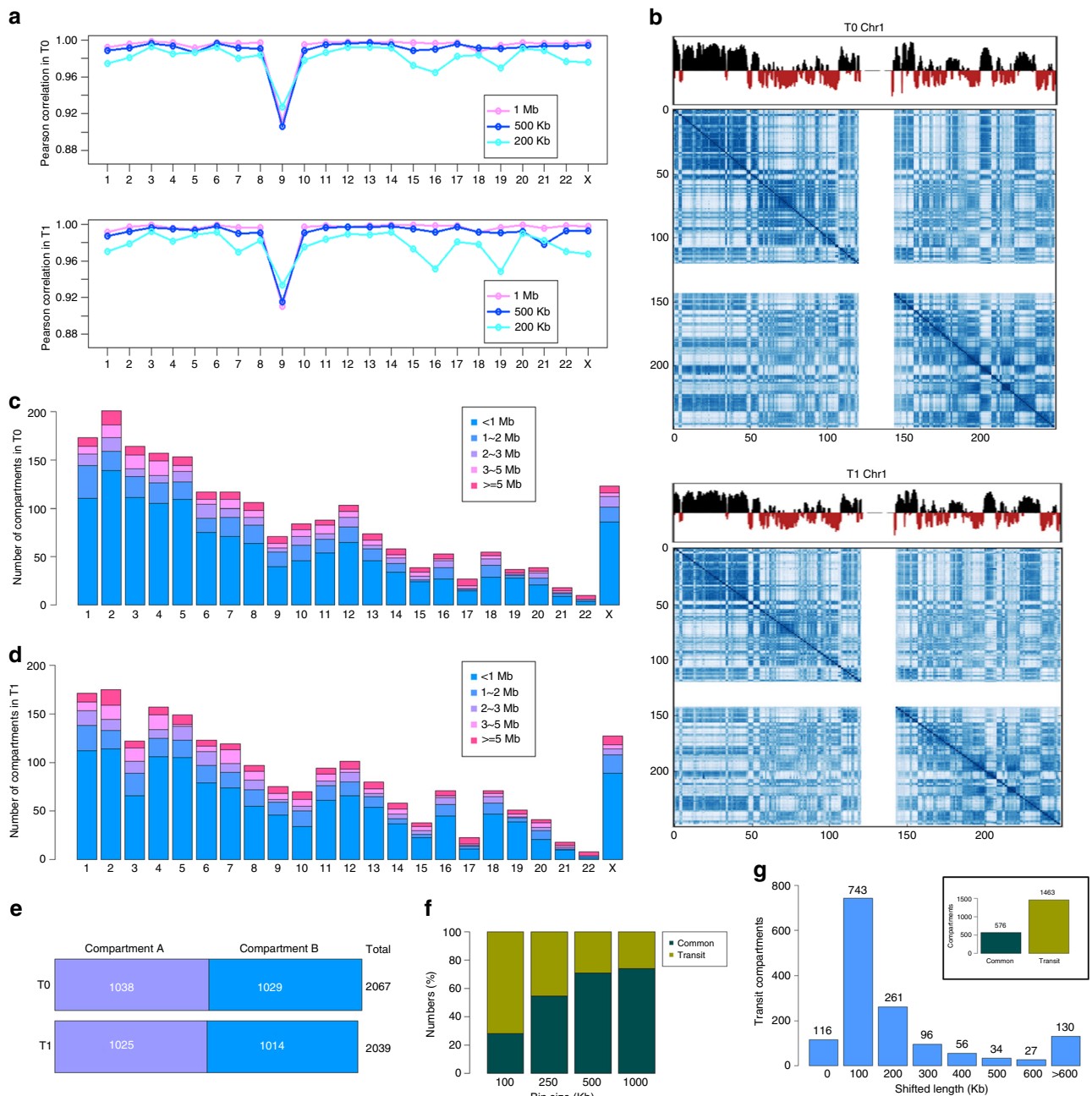

**Fig. 1** Identification of E2-induced compartments in MCF7 cells at T1 vs. T0. **a** Pearson correlation of two biological replicates with the bin size of 1 Mb, 500 Kb and 200 Kb on individual chromosomes at untreated MCF7 cells (T0) (Top) and E2-treated MCF7 cells for 1 h (T1) (Bottom). **b** Contacts matrices of compartment A (Black) and B (Red) at T0 (Top) and T1 (Bottom) respectively. **c**, **d** Histograms of compartments with different sizes at T0 and T1 respectively. **e** Distribution of compartments A and B in T0 and T1 respectively. **f** The percentage of common and transit compartments at T1 vs. T0 with various bin sizes. **g** The number of compartments of T1 vs. T0 with various shifted length when bin size is 100 Kb

**Temporal dynamic chromatin along prolonged E2 treatments.** Several studies found transcriptional response to longer E2 treatment was dramatically different from early or shorter E2 treatment[30,31]. In order to understand the dynamics of chromatin structure in longer E2 treatment periods in MCF7 cells, we further conducted TCC analysis in three more time points, 4 h (T4), 16 h (T16) and 24 h (T24), each with biological replicates (Supplementary Table 1). In order to capture the dynamic patterns following a prolonged E2 treatment, we determined compartments at 100 Kb and compared the compartments among the five time points. As expected, the number of compartments was very similar among the five time points (Supplementary Table 2,

Supplementary Figs. 6–11). We used the Common and Transit compartments obtained from the comparison of T1 vs. T0 and compared them to T4, T16 and T24 respectively. We also separated the comparison of active open chromatin compartment A (Fig. 2a—upper panel) from inactive close chromatin compartment B (Fig. 2a—lower panel). When re-examining these sets of Common or Transit compartments, we identified 15 patterns of changed compartments from 16 sets (Supplementary Fig. 14) and 9 additional patterns from the last set (labeled X in Supplementary Fig. 14) based on the converted bins (see the definition in Supplementary Fig. 15). Unexpectedly, we were able to categorize these 24 patterns of chromatin into six types of temporally

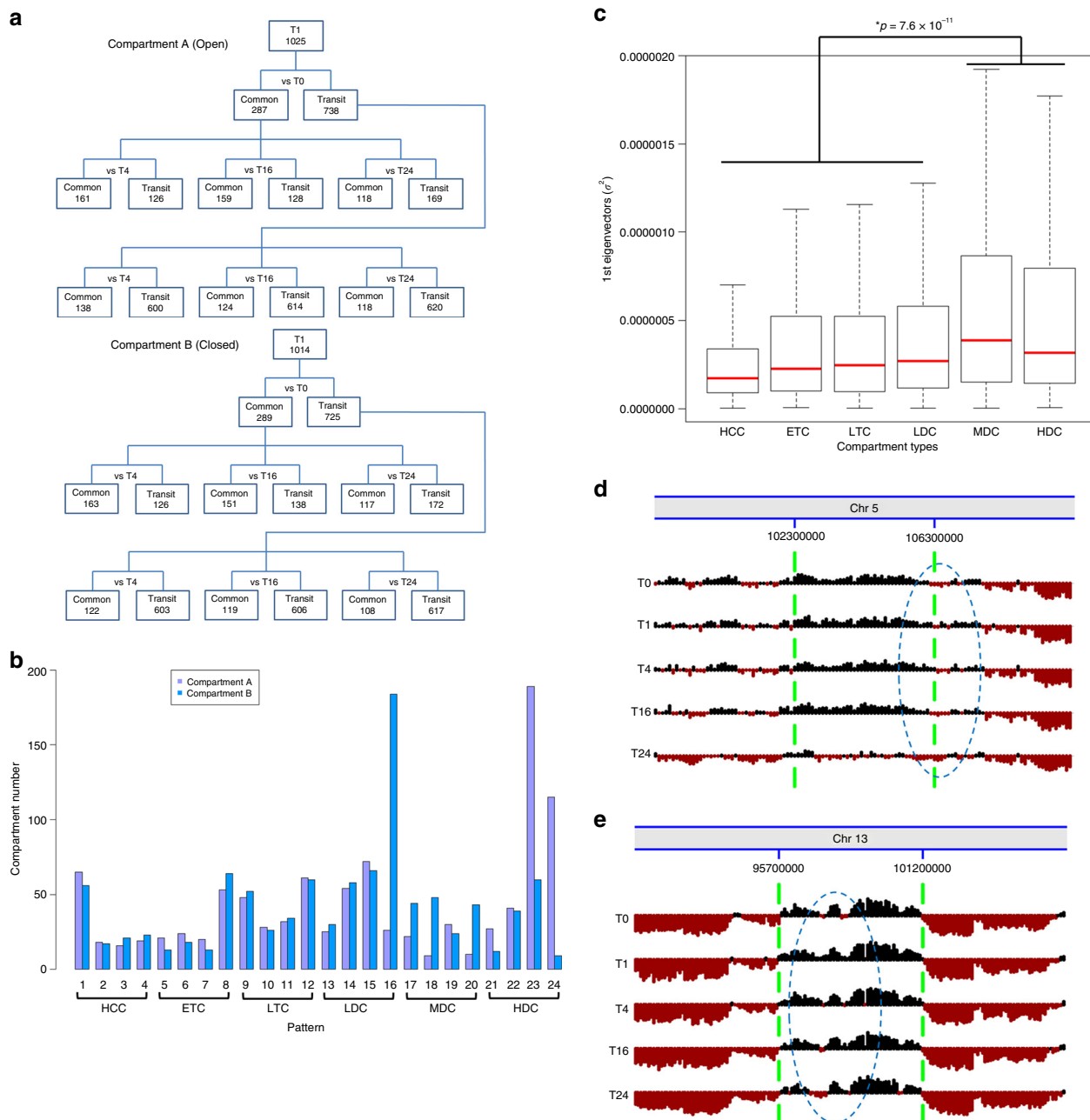

**Fig. 2** Determination of E2-induced TDRCs in MCF7 cells. **a** An instructive tree of comparing compartments A (Top) and B (Bottom) at E2-induced five time points. T0: control, T1: 1 h; T4: 4 h; T16: 16 h; and T24: 24 h. **b** A number of compartments at each of 24 individual patterns categorized into six types, including HCC with patterns 1~4; ETC with patterns 5~8; LTC with patterns 9~12; LDC with patterns 13~16; MDC with patterns 17~20; HDC with patterns 21~24. **c** The variance ($\sigma^2$) of first eigenvector values of compartment types in T0, T1, T4, T16, T24. The p value was determined by Wilcoxon rank-sum test. **d**, **e** Examples of compartment changes, MDC (**d**) and HDC (**e**) along E2-induced five time points. Dark: Compartment A; Dark red: Compartment B. Compartments are indicated between green dash lines. Blue dash ovals highlight the dynamic region. TDRC temporally dynamic re-compartmentalization, MDC moderately dynamic compartments, HDC highly dynamic compartments

dynamic re-compartmentalization (TDRC): highly common compartments (HCC, patterns 1−4 with an false discovery rate (FDR) of 0.268), early transit compartments (ETC, patterns 5−8 with an FDR of 0.230), late transit compartments (LTC, patterns 9−12 with an FDR of 0.192), lowly dynamic compartments (LDC, patterns 13−16 with an FDR of 0.178), moderately dynamic compartments (MDC, patterns 17−20 with an FDR of 0.161), and highly dynamic compartments (HDC, patterns 21−24 with an FDR of 0.201) (Fig. 2b). There is also a statistically

significant difference ($p = 7.6 \times 10^{-11}$, Wilcoxon rank-sum test) between highly dynamic compartments (MDC and HDC) and lowly dynamic compartments (HCC, ETC, LTC and LDC) (Fig. 2c). Interestingly, the MDC and HDC are predominantly composed of compartment A mainly from Transit to Transit compartments along the time courses of E2 treatment (Fig. 2d, e), while the LDC has the most changed compartment B. These data suggest that active chromatin domains are more susceptible to change in response to E2 stimulation over time.

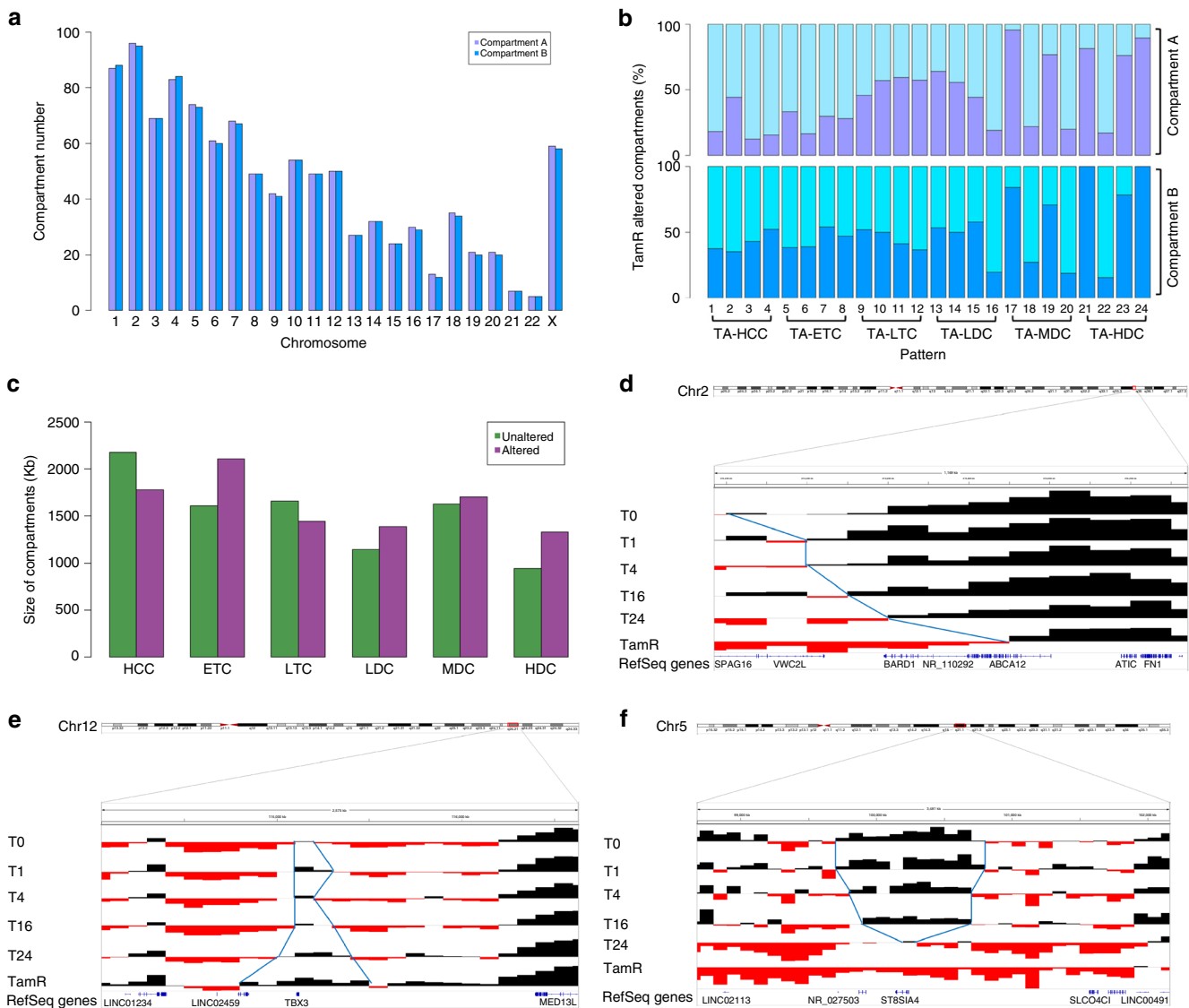

**Fig. 3** Identification of TRACs in TamR cells. **a** Number of tamoxifen-resistant compartments. **b** The percentage of altered compartments, A (Top panel) and B (Bottom panel) in TamR cells. **c** Genomic size of six types of TRACs and TRUCs. **d**−**f** UCSC genome browser snapshots of TRACs and the enclosed genes within their loci. Dark: Compartment A; Red: Compartment B. **d** Shrunk compartment. Blue lines represent the compartment boundary. **e** Expanded compartment. **f** Flipped compartment. TRAC tamoxifen-resistant altered compartment, TRUC TamR unaltered compartment

**Altered chromatin compartmentalization in resistant cells.** Increasing evidence suggests that ERα-mediated gene deregulation or epigenetic alterations may be key mechanisms underlying acquired tamoxifen-resistant breast cancer[42,43]. However, the knowledge of endocrine-resistant associated 3D regulation is still limited. To delineate the altered 3D architectures, we further conducted TCC analysis in a tamoxifen-resistant MCF7 cell line, MCF7-TamR[44] (Supplementary Table 1, Supplementary Figs. 12, 13). At a resolution of 100 Kb, we identified 2103 compartments including both A and B (Fig. 3a). We further defined a compartment to be a tamoxifen-resistant altered compartment (TRAC) if it was a Transit compartment and if there was at least one converted bin between TDRC and a TamR compartment. As such, we obtained six corresponding types of TRACs: TA-HCC (FDR of 0.154), TA-ETC (FDR of 0.250), TA-LTC (FDR of 0.154), TA-LDC (FDR of 0.165), TA-MDC (FDR of 0.139) and TA-HDC (FDR of 0.226) (Supplementary Data 1). Clearly, patterns 17, 19, 21, 23, and 24 in TA-MDC and TA-HDC types showed higher alteration than other patterns, suggesting that the

higher dynamics of the compartments in E2-induced MCF7 cells, the stronger alterations of the compartments in TamR cells (Fig. 3b). Further, we observed that the average size of TA-HDC and TA-ETC types was longer than those in unaltered compartments, while the size of TamR unaltered compartments (TRUCs) was longer than TA-HCC (Fig. 3c). By a bird's-eye-view examination, we identified three interesting types of TRACs: Shrunk, Expanded and Flipped (Fig. 3d−f). Our data suggest that a group of genes within the same domain may be concordantly regulated during acquired tamoxifen resistance.

**Epigenetic states in dynamic recompartmentalization.** Epigenetic marks have been shown to classify genomic compartments and chromosomal domains into subcompartments or subdomains in diverse cell types[18]. There is little known about the structural roles of one-dimensional (1D) epigenetic states in E2-induced 3D chromatin structure. We performed ChIP-seq of three active marks, H3K27ac, H3K4me1, and H3K4me3, and two repressive marks, H3K9me3 and H3K27me3 in a time

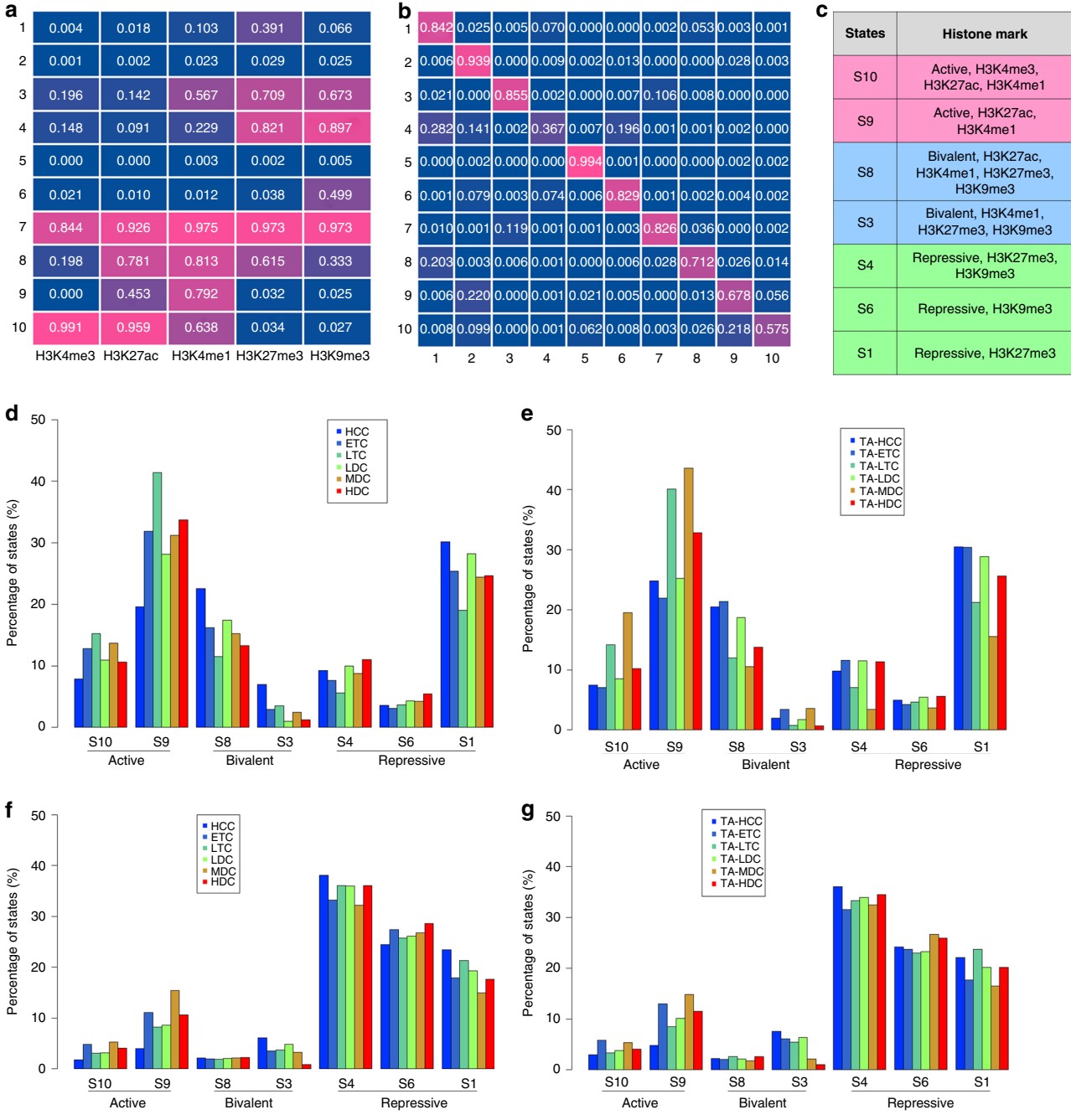

**Fig. 4** Epigenetic modifications on E2-induced TDRCs and TRACs. **a** Emission probabilities of ten epigenetic states trained by an HMM model on five histones. The enrichment of the corresponding mark indicated by high values and dark red color. **b** Transition probabilities of ten epigenetic states trained by an HMM model on five histones. The enrichment of the corresponding mark indicated by high values and dark red color. **c** The summary of the corresponding histone marks to each of biologically meaning epigenetic states. **d** The percentage of epigenetic states on dynamic E2-induced compartment A. **e** The percentage of epigenetic states on altered TamR compartment A. **f** The percentage of epigenetic states on dynamic E2-induced compartment B. **g** The percentage of epigenetic states on altered TamR compartment B. TDRC temporally dynamic re-compartmentalization, TRAC tamoxifen-resistant altered compartment

course of E2 treatment in MCF7 cells as well as in asynchronous TamR cells, each with biological replicates (Supplementary Table 3). We first trained a total of 30 histone modification data by ChromHMM at various parameters[45], and obtained ten HMM states (Fig. 4a, b). Through interpreting both HMM emission and transition probabilities, we inferred seven biologically meaningful epigenetic states, including two active states (S9 and S10), two bivalent states (S3 and S8) and three repressive states (S1, S4 and S6) (Fig. 4c). We were also able to map

these states back into each of six types of E2-induced dynamic compartments and TamR altered compartments. Overall, we observed that more active states were distributed in compartment A and more repressive states in compartment B, while three types of dynamic changed compartments, i.e., LDC, MDC and HDC, have a higher percentage of active states than HCC does (Fig. 4d−g). Surprisingly, S1 showed a high percentage of distribution in compartment A despite that it is a repressive state.

**ERα and CTCF binding in dynamic recompartmentalization**. Since ERα is a master TF mainly in response to E2 stimulation in MCF7 cells and CTCF is a chromatin organizer known to regulate the 3D architecture, we wanted to understand their regulatory roles in mediating these 3D structural dynamics. We performed ChIP-seq of ERα and CTCF at five time points in E2-stimulated MCF7 cells and in TamR cells, each with biological replicates (Supplementary Table 3). We used MACS[46] to call ERα binding sites (peaks) in each of the 12 data sets and obtained 7553 peaks in untreated MCF7 cells (T0), and between 13,000 and 20,000 ERα peaks in E2-treated MCF7 cells at four time points and untreated TamR cells respectively (Fig. 5a). Interestingly, we found Patterns 18 and 20 of MDC and Pattern 24 of HDC had the highest number of ERα binding sites in compartment A but not in compartment B in E2-induced MCF7 cells (Supplementary

Fig. 17–19) as well as in TamR cells (Fig. 5b), illustrating that there are more ERα binding sites in higher dynamic active chromatin.

Furthermore, we identified ~50,000 CTCF peaks in each of the five time points in E2-induced MCF7 cells and in TamR cells (Fig. 5c). On the switched domain boundary between two compartments, we observed generally lower averages of number of CTCF binding sites in three types of TDRCs (LDC, MDC, HDC), and three types of TRACs (TA-LDC, TA-MDC, TA-HDC) than other three types (Fig. 5d). When testing the correlation of ERα binding within the compartments vs. CTCF binding on the boundary regions, Pattern 24 of HDC or TA-HDC was the only pattern having more than 80% ERα peaks and less than 0.4 CTCF peaks per compartment in E2-induced MCF7 cells (Fig. 5e—left panel) or in TamR cells (Fig. 5e—right panel).

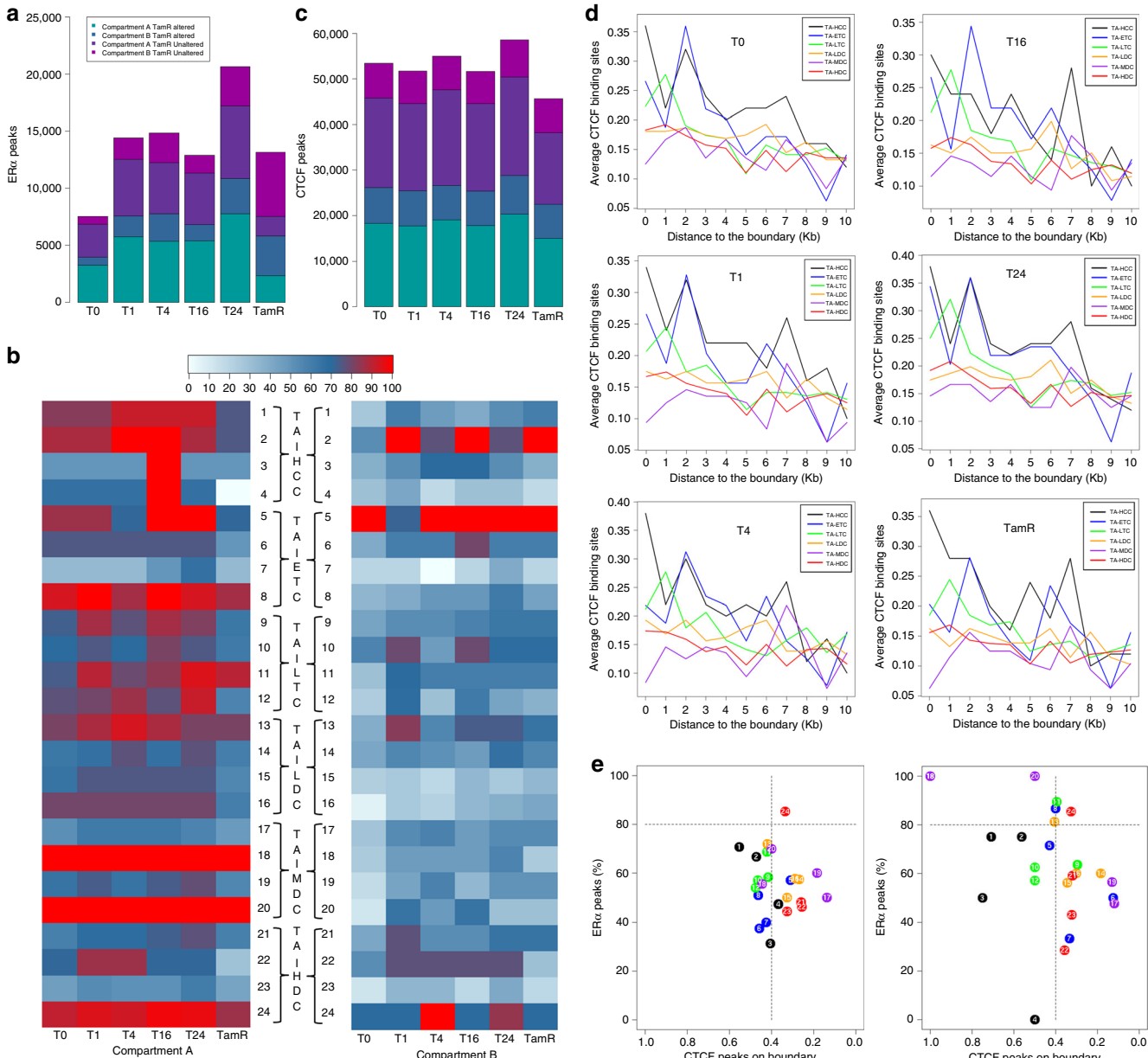

**Fig. 5** A distribution of ERα and CTCF peaks in E2-induced TDRCs and TRACs. **a** The number of ERα peaks. **b** Heatmap showing the percentage of altered TamR compartments A (Left) and B (Right) with ERα peaks. **c** The number of CTCF peaks. The legend is the same as (**a**). **d** The average of CTCF peaks on altered TamR compartment boundary in E2-induced five time points and TamR MCF cell lines. **e** The distribution of each of 24 patterns (circled) of ERα peaks within compartments and CTCF peaks on compartment boundary in E2-induced MCF7 cells (left) and in TamR cells (right). CTCF CCCTC-binding factor, TDRC temporally dynamic re-compartmentalization, TRAC tamoxifen-resistant altered compartment

Collectively, our results indicate a reciprocal relationship to ERα binding and CTCF binding at highly dynamic changed compartments during the temporal response to E2 stimulation which is also observed in cells with acquired tamoxifen resistance.

**Differentially expressed genes and putative loops.** We next examined the gene expression and loops in these TDRCs and TRACs. We utilized a publicly available time-series RNA-seq data[47], which was profiled at ten time points of E2-treated MCF7 cells, to identify differentially expressed genes. By picking up five time points close to this study (T0′ = 0 min, T1′ = 40 min, T4′ = 160 min, T16′ = 640 min, T24′ = 1280 min), we identified 4106 dynamic differentially expressed genes (DDEGs) across all time points. As expected, a majority of such genes were located in dynamically changed compartments A with most in the MDC and HDC types (Fig. 6a). There seems to be minimal difference in the average of gene expression levels within each of six types of the TDRCs. However, the variance of DDEGs in LDC, MDC and HDC is higher than HCC, ETC and LTC (Fig. 6b). GSEA analysis[48] showed ribosome, tight junction, endocytosis, lysosome, cell cycle, WNT signaling pathway, insulin signaling pathway, focal adhesion, and MAPK signaling pathway were among the top functional categories for 1396 DDEGs in MDC and HDC types (Fig. 6c, Supplementary Data 2).

We further performed RNA-seq in parental MCF7 and TamR cells, each with three biological replicates (Supplementary Table 5), and identified a total of 2097 TamR-specific differentially expressed genes (TDEGs). More than half of them (1188) were in the combined TA-M&HDC types (Fig. 6d, Supplementary Data 3). We then identified 42,390 TamR-specific significant interaction pairs or putative loops from TamR TCC data by HOMER[49] and using T0 TCC data as the contrast. 3C-qPCR validations further confirmed the differential looping intensity of seven randomly selected pairs between parental MCF7 and TamR cells (Supplementary Fig. 20 and Supplementary Table 5). Of 42,390 identified loops, 16,807 were overlapped with a promoter (−5 K/+1 K around 5′TSS), 9638 of them had either H3K27ac or H3K4me1 peaks in the distal loci, 4122 of them had at least one ERα binding site at either loci of the loop and were thus considered as ERα-regulated promoter−enhancer (ERα-PE) loops (Fig. 6e). Finally, 396 TDEGs overlapped with 599 ERα-PE loops in the combined TA-M&HDC types were identified as ERα dysregulated dynamic looping genes in resistant cells (Supplementary Data 4-5). Functional annotation and gene pathways with GSEA showed they were enriched with nine KEGG signaling pathways related to cancer invasion and aggressiveness, as well as glycolysis and metabolism (Fig. 6f, Supplementary Figs. 21–28), indicating the property of resistant cells may resemble advanced cancer cell types. Taken together, our results demonstrated that these ERα-associated dynamically reorganized active domains in regulating gene looping events may result in higher susceptibility to alterations in tamoxifen-resistant cells. This prompts us to speculate that these genome domains and looping genes may be responsible for driving the acquired tamoxifen resistance.

## Discussion

Despite the increasing developments of various 3C-derived high throughput sequencing techniques in which it advances our understanding of the principles of 3D genome architecture, several important questions remain to be answered in the field. One of many attempts is to elucidate how stable or dynamic chromosome domains are upon signaling stimuli and to what extent these changes affect establishing or re-establishing the compartmentalized architecture. Our main goal of this study is to establish a basis for data-driven modeling of temporal dynamics and 3D chromatin reorganization given that such studies are very limited. While mega-sized TADs are conserved among different cell types and mammalian species[3,5,12], 100−500 Kb size of subTADs or compartments are considered to be dynamic or the boundaries are nonconserved[18,50]. Though our data showed the number of compartment domains are quite similar among different time points, the changes in size (at least 100 Kb) of compartments are very pronounced, particularly in these E2-induced highly dynamically changed compartments. With a very loose definition of dynamic changes requiring a minimum of 100 Kb, we were able to unveil 24 temporal dynamic patterns upon E2-induction which were further categorized into six major types. Indeed, the MDCs and HDCs were not only predominately of active compartment A but also enclosed dynamic differentially expressed genes enriched with biological process terms ribosome, tight junction, cell cycle and others (Fig. 6c). Many of them characterize known effects of estrogen on MCF7 cell phenotype[51]. In contrast, there were no significant differences between E2-induced early and late responded changed compartments, in which both types were comprised of very few compartments. Our data implied that these moderate to higher dynamically changed compartments may play an essential role in governing this hormone-mediated luminal breast cancer development.

Many studies including ours have demonstrated that E2 instructed dynamic transcriptional program rewired or altered transcription regulatory networks in tamoxifen-resistant breast cancer cells[36,52,53]; however, very few were focused on examining 3D regulatory roles in tamoxifen resistance. Our previous studies utilized a 3C-seq technique to identify two densely mapped DERE regions located on chromosomes 17q23 and 20q13 frequently amplified in MCF7 cells and found their aberrantly amplified DEREs deregulated target genes were potentially linked to cancer development and tamoxifen resistance[38]. However, there are many limitations in that study, including the technique itself, smaller data volumes and fewer computational tools available for a thorough analysis. Our current work has significantly improved in the following aspects: (1) generating a high quality and a high sequencing depth of TCC data, such depth can at least capture a 40 Kb resolution; (2) producing E2-induced time series of TCC in MCF7 cells and then comparing it to TamR cells; and (3) utilizing many state-of-art computational tools in processing TCC and ChIP-seq data. Remarkably, our integrative analyses uncovered many temporal dynamic patterns characterizing the 3D chromatin reorganization upon E2-induction. Interestingly, two types of temporal dynamic re-compartmentalization (TDRC), i.e., moderately and highly dynamic compartments (MDCs and HDCs), showed higher alteration in TamR cells (Fig. 3b). Furthermore, the looping gene signatures enclosed in these two altered dynamic domains were highly enriched with GO terms cancer invasion and aggressiveness or metabolism. All of these biological processes captured the nature of acquired resistant breast cancer cells[54].

Although our definition of 24 patterns or six types of dynamic changed compartments were based upon how recompartmentalization in MCF7 cells respond to E2 induction, interestingly, the resultant six types are identical to the analysis based on a mathematical calculation resulting in a total of 256 combinations of E2-induced time-dependent compartments (Supplementary Fig. 29) when using T0 as a contrast. Further, we observed the same trends of E2-induced time-dependent compartments in both MCF7 and T47D cell lines (Supplementary Fig. 30), where the major trend is miscellaneous (dynamic changed) compartments in both MCF7 and T47D cell lines. Moreover, the altered compartments of

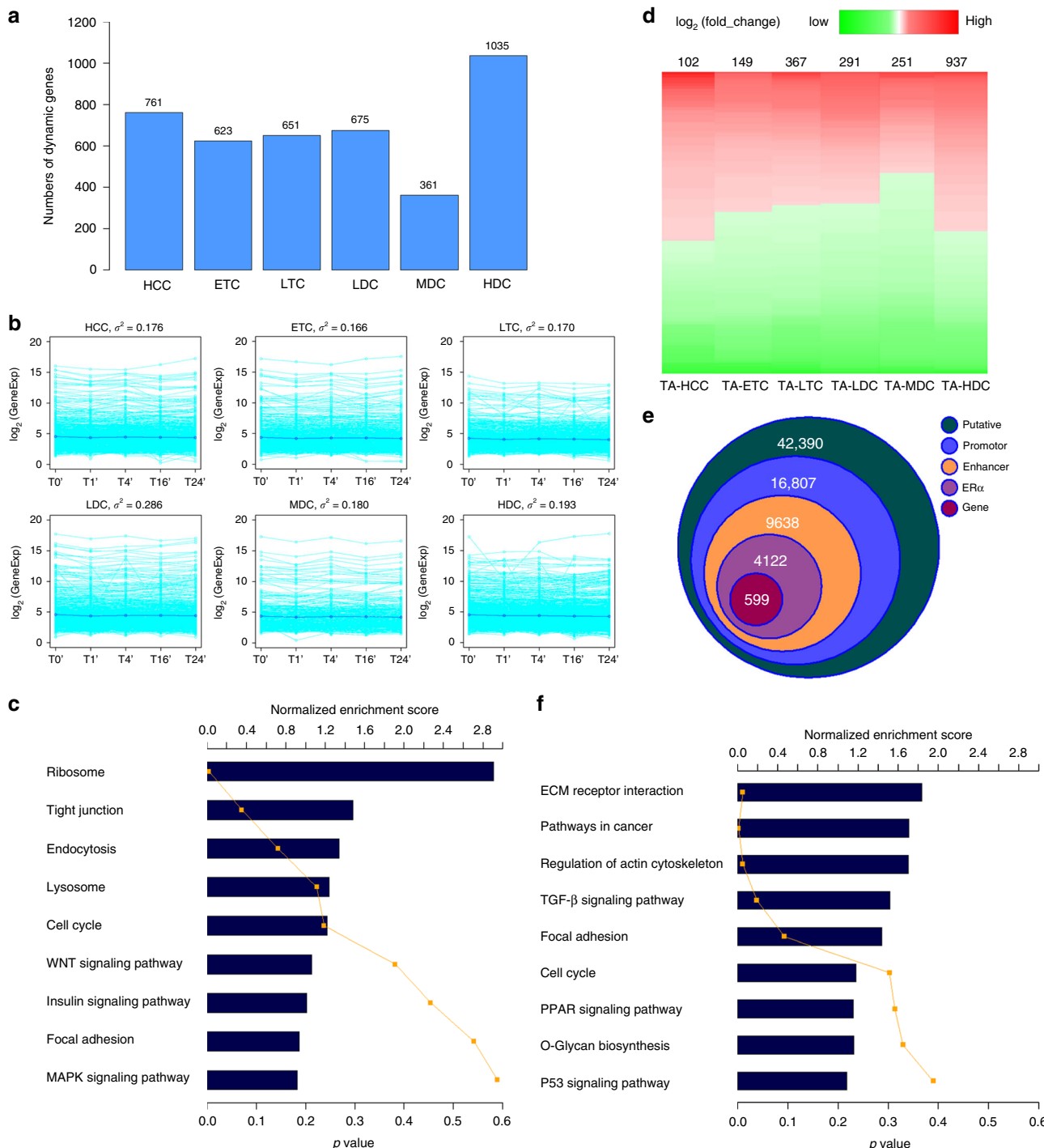

**Fig. 6** Gene expression and looping in E2-induced TDRCs and TRACs. **a** The number of DDEGs located in each of six types of TDRCs. **b** A plot of gene expression values along five time points in each of six types of TDRCs. Each green line represents one gene, and blue line represents the average gene expression value. $\sigma^2$ is the variance of $\log_2$ values of gene expression. **c** The enrichment of KEGG pathways of 1396 genes in MDC/HDCs. **d** The heatmap of DEGs of TamR vs. MCF7 in each of six types of TRACs. **e** The number of loops defined by HOMER. Putative: putative loops identified by HOMER. Promoter: one locus of putative loops within −5 K/+1 K of TSS. Enhancer: one locus of putative loops within −5 K/+1 K of TSS and the other with either H3K27ac/ H3K4me1 peaks. ERα: at least one locus of loops with an ERα peak and promoter−enhancer. Gene: genes associated with ERα-PE loops showing differentially expressed at TA-MDCs and TA-HDCs. **f** The enrichment of KEGG pathways of 396 genes associated with 599 ERα-PE loops. TDRC temporally dynamic re-compartmentalization, TRAC tamoxifen-resistant altered compartment, DDEG dynamic differentially expressed genes, KEGG Kyoto Encyclopedia of Genes and Genomes, MDC moderately dynamic compartments, HDC highly dynamic compartments

both MCF7-TamR and T47D-TamR have higher percentage of miscellaneous compartments (Supplementary Figs. 31, 32). Our results suggest that our analytical strategy and observations are generalizable in various cell lines.

Our findings further illustrated an anticorrelative trend of binding enrichments between intradomain ERα sites and boundary CTCF sites (Fig. 4d, e). Interestingly, the average of CTCF sites is generally lower regardless of its distance from the

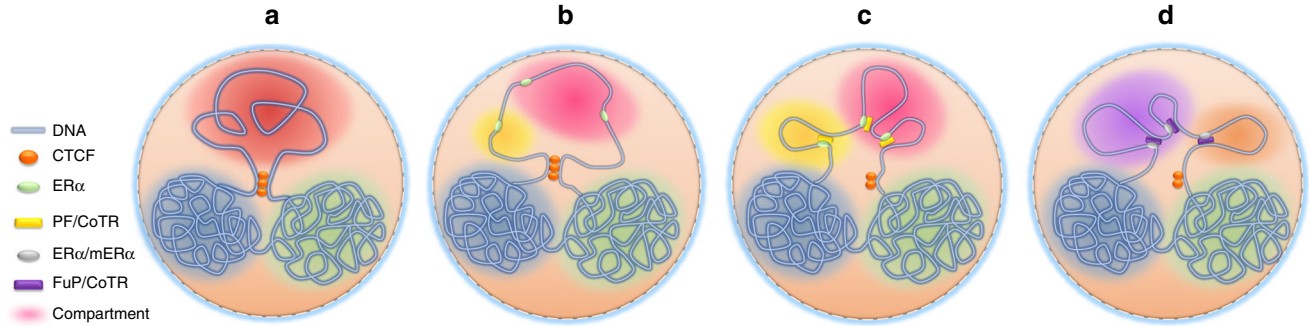

**Fig. 7** A proposed model for dynamic 3D chromatin architecture. **a−c** Constitutive estrogen stimulation in breast cancer cells enhances stronger ERα activity and further recruits its distal regulatory machinery and then mobilizes highly dynamic gene looping which essentially expand to render a 3D genome re-compartmentalization accompanying with lower CTCF binding at the compartment boundary. **d**. In acquired resistant breast cancer cells, increased crosstalk between ERα and other signal transduction pathways such as EGFR/HER2, IGF-IR, and AKT/PTEN or altered expression of some key coregulators particularly reshuffle these highly dynamic gene looping resulting in altered chromatin reorganization. CTCF CCCTC-binding factor

boundary in these highly dynamic changed domains. Although our result is not so surprising, it nevertheless implicates an underlying molecular event that ERα-regulated highly recompartmentalization may be required to loosen CTCF insulator at the domain boundary[4,55]. Furthermore, our data may support the high-order chromatin architectural role of ERα signaling in mediating hormonal activity, expanding our current understanding of the molecular mechanism underlying the E2-induced breast carcinogenesis through ERα regulation.

Collectively, our observations suggest a possible molecular mechanistic model (Fig. 7a−d). A constitutive estrogen stimulation in breast cancer cells enhances stronger ERα activity and further recruits its distal regulatory machinery including different coregulators, mediators, cohesions and chromatin remodelers, and then mobilizes highly dynamic gene looping events which essentially expand to render a 3D genome recompartmentalization accompanied with lower CTCF binding at the compartment boundary. In contrast, in breast cancer cells with the acquired resistance, increased crosstalk between ERα and other signal transduction pathways such as EGFR/HER2[56], IGF-IR[53], and AKT/PTEN[57] or altered expression of some key coregulators particularly reshuffle these highly dynamic gene looping resulting in altered chromatin reorganization. To substantiate this model, we propose further functional or mechanistic experiments in our follow-up studies: (1) establishing genome-edited TamR sublines by editing out a handful ERα sites selected from 599 gene loops using the CRISPR/Cas9 technique[58]; in these sublines, examining the changes of ERα-regulated looping using ChIP-3C-qPCR and determining if resistant cells are re-sensitized; and (2) establishing an in vivo model of TamR xenografts[44]; in this model, examining selected ERα-regulated loops using ChIP-3C-qPCR by comparing untreated vs. treated Gefitinib, an inhibitor to EGFR/HER2. With these results, we might expect to elucidate the detailed 3D ERα regulatory role in mediating tamoxifen resistance.

In summary, the high-quality and large 3D chromatin data along with many ChIP-seq and RNA-seq data provided a comprehensive resource for understanding how estrogen exposure drives genome-wide 3D chromatin reorganization in ERα-positive breast cancer cells as well as how their alterations occur in hormone resistant cells. Our integrative analysis reveals temporal dynamic patterning and 3D chromatin reorganization of the breast cancer genome that occurs in response to E2 stimulation over time. Our work may give further insight into the effective treatment strategies to overcome tamoxifen resistance and discovery of novel epigenetically therapeutic targets.

## Methods

**Cell lines and reagents**. The human parental MCF7, T47D and Tamoxifen-resistant (TamR) cell lines were derived from Osborne et al.[54]. MCF7 cells were cultured in Dulbecco's modified Eagle's medium (DMEM) supplemented with 10% fetal bovine serum (FBS), 2 mM L-glutamine and 1% penicillin/streptomycin (pen/strep) until 90% confluent. For temporal estrogen responsiveness, MCF7 cells were hormone-starved for 72 h followed by the addition of 100 nM β-Estradiol (MP Biomedicals, Inc.) at 1, 4, 16 and 24 h. To hormone-starve MCF7 or T47D cells, these cells were grown to 80% confluency as described above. Once the desired confluency was reached, the cells were washed one time with phosphate-buffered saline and the media was replaced with phenol-red free DMEM supplemented with 5% charcoal-stripped FBS, 2 mM (L-glutamine) and 1% (pen/strep). For the 0 h time point, cells were immediately crosslinked following 72 h of hormone starvation.

TamR cells were cultured in phenol-red free DMEM supplemented with 10% charcoal-stripped FBS, 2 mM L-glutamine, 1% pen/strep, and 100 nM Tamoxifen (Sigma-Aldrich). Tamoxifen was replenished every 48 h and cells were crosslinked at 90% confluency.

**Tethered chromatin capture (TCC)**. TCC was performed as Kalhor et al. described[40]. Approximately 50 million MCF7 or T47D cells (either parental hormone-starved with temporal addition of β-Estradiol (E2) or TamR cells as described above) were crosslinked with 1% formaldehyde for 10 min at room temperature, crosslinking was quenched with 0.125 M glycine for 5 min at room temperature and cell pellets were collected and stored at −80 °C. The crosslinked cells were lysed with 550 μL of Cell Lysis Buffer (10 mM HEPES pH 8.0, 10 mM NaCl, 0.2% Igepal CA-630, containing 1× protease inhibitor cocktail (Thermo Scientific #88665) and 1× PMSF (Acros Organics #215740050). The cells were homogenized with a dounce homogenizer for 20 strokes with pestle A after incubation on ice for 15 min. The lysate was then centrifuged at 2500 rcf for 5 min at room temperature. The supernatant was discarded and the remaining pellet containing the nuclei was washed twice with ice-cold wash buffer #1 (50 mM Tris-HCl pH8, 50 mM NaCl, 1 mM EDTA) and resuspended in 250 μL of wash buffer #1. The chromatin was then solubilized by the addition of 95 μL of 2% sodium dodecyl sulfate (SDS) followed by an incubation at 65 °C for 10 min. The cysteine residues were biotinylated by the addition of 105 μL of 25 mM EZlink Iodoacetyl-PEG2-Biotin (IPB) (Thermo Scientific #21334) and incubated at room temperature for 1 h while rocking. The SDS was neutralized with 1300 μL of 1× NEBuffer 2 (NEB #B7002S) on ice for 5 min, followed by the addition of 225 μL of 10% Triton X-100, which was then incubated on ice for 10 min followed by a final incubation at 37 °C for 10 min. DNA was digested overnight at 37 °C with the following reagents: 100 μL of 10× NEBuffer 2, 5 μL of 1 M DTT, 430 μL of water, and 2000U of HindIII (100 U μL⁻¹; NEB #R0104M). To remove remaining IPB, the samples were then dialyzed for 4 h at room temperature using a Slide-A-Lyzer Dialysis Cassette with a 20 kD cutoff (Thermo Scientific #87735) in 1 L of dialysis buffer (10 mM Tris-HCl, pH 8.0 and 1 mM EDTA). The biotinylated chromatin was then tethered using 400 μL of MyOne Streptavidin T1 beads (Invitrogen #65601) after washing the beads three times with PBST (1× PBS containing 0.01% Tween20) and resuspending in 2 mL of PBST. Four hundred microliters of washed Streptavidin T1 beads was then added into each of five equal aliquots of dialyzed sample. Binding occurred at room temperature for 30 min followed by the addition of 150 μL of 25 mM IPB neutralized with 25 mM 2-mercaptoethanol, which was then incubated at room temperature for 15 min. Non-crosslinked DNA and nonbiotinylated chromatin was removed by washing the beads once with 600 μL PBST followed by one wash with 600 μL wash buffer #2 (10 mM Tris-HCl, pH 8.0, 50 mM NaCl, 0.4% Triton X-100). To wash the beads, we utilized a magnetic rack and ensured beads bound to the magnet before aspirating the buffer out. The beads

were then resuspended in 100 μL of wash buffer #2. The 5′ overhangs were filled with 63 μL water, 1 μL 1 M MgCl₂, 10 μL 10× NEBuffer 2, 0.7 μL 10 mM dATP (NEB #N0440S), 0.7 μL 10 mM dTTP (NEB #N0443S), 0.7 μL 10 mM 2′-Deoxyguanosine-5′-O(1-thiotriphosphate) sodium salt (dGTPαS) (Biolog Life Science Institute #D031-05), 15 μL of 0.4 mM Biotin-14-dCTP (Invitrogen #19518-018), 4 μL of 10% Triton X-100, and 25U Klenow-large fragment (NEB #M0210L) and rocked at room temperature for 40 min. Five microliters of 0.5 M EDTA was added to stop the reaction and the beads were washed twice with wash buffer #3 (50 mM Tris-HCl pH7.4, 0.4% Triton X-100, 0.1 mM EDTA) and resuspended in 500 μL of wash buffer #3. The crosslinks were reversed with 400 μL of extraction buffer (50 mM Tris-HCl pH8, 0.2% SDS, 1 mM EDTA, 100 mM NaCl) followed by the addition of 400 μg of proteinase K (NEB #P8107S) and incubation for 2 h at 65 °C. The initial conformation capture library (the supernatant) was extracted twice with an equal volume of phenol:chloroform:isoamyl alcohol (25:24:1 v/v) and once with an equal volume of chloroform. NaCl was then added to a final concentration of 20 mM and glycogen to 2 μg μL⁻¹ followed by precipitation of the DNA with the addition of 900 μL of ethanol (200 proof) and incubation at −20 °C overnight. The DNA was pelleted via centrifugation at 20,000 rcf at 4 °C for 20 min. The pellet was then immersed in 500 μL of 80% ethanol and centrifuged at 20,000 rcf for 10 min. The ethanol was removed and the pellet was air dried until approximately 90% dry and resuspended in 20 μL of 10 mM Tris-HCl pH8. The five aliquots were combined and the RNA was removed via RNAseA digestion (10 μg RNAseA) for 30 min at 37 °C. The DNA was purified using the Invitrogen Purelink Quick PCR purification kit (Invitrogen #K310001). Biotin from nonligated DNA was removed from 5 μg of purified DNA using 300U EXOIII (NEB #M0206S), adjusting the total volume to 90 μL with 10× NEBuffer 1 (NEB #B7001S). This reaction was incubated at 37 °C for 1 h. The reaction was stopped with 2 μL of 0.5 M EDTA and 2 μL of 5 M NaCl followed by incubation at 70 °C for 20 min. A Covaris Focused-ultrasonicator (Covaris S220) was used to shear the DNA, with a duty factor of 5%, peak power of 175 W, and 200 cycles per burst. Each sample was sonicated for 180 s and purified using the Purelink Quick PCR purification kit and eluted in 50 μL of elution buffer. Libraries were generated with the NEBNext Ultra II DNA Library Prep Kit for Illumina (NEB #E7645L). First, end-repair was performed after sonication. One microgram of DNA was used and the total volume of sample was brought up to 50 μL with 0.1× TE. The end repair was carried out as outlined in the manufacturer's protocol. After end repair, the biotinylated DNA was pulled down using 10 μL of MyOne Streptavidin C1 beads (Invitrogen #65001). The beads were first washed twice with 500 μL of 1× Binding and Wash buffer (for 2× Binding and Wash buffer: 10 mM Tris-HCl pH7.5, 1 mM EDTA, and 2 M NaCl) and resuspended in 2× Binding and Wash buffer, which was then added to the end-repaired DNA. Lo-bind tubes (Eppendorf #022431021) were used to prevent sticking of beads to the sides of the tubes. The samples were rocked at room temperature. The beads were washed one time with 1× Binding and Wash buffer containing 0.1% Triton-X-100 followed by one wash with 10 mM Tris-HCl, pH8 and the beads were collected in 60 μL of 10 mM Tris-HCl, pH8. Next, adaptor ligation was performed as described in the NEBNext Ultra II DNA Library Prep protocol, following the instructions for 1 μg of input DNA. After ligation of Illumina adaptors on the beads, the beads were washed twice with 1× Binding and Wash buffer and twice with 0.1× TE. The beads were resuspended in 30 μL of 10 mM Tris-HCl pH8. Fifteen microliters of the beads containing adaptor-ligated DNA was transferred to a new tube and we continued on to PCR enrichment of adaptor-ligated DNA on the beads. The remaining 15 μL was saved and stored at 4 °C. The PCR was carried out as outlined in the NEBNext Ultra II protocol, NEBNext Multiplex oligos for Illumina (NEB #E7335S and #E7500S) were used for the individual barcodes and the enrichment was performed using ten cycles. Forty-five microliters of the supernatant containing the PCR products was transferred to a new tube and were cleaned using a 0.8× bead cleanup of the PCR reaction with Agencourt AMPure XP beads (Beckman Coulter #A63881). The AMPure XP beads were brought to room temperature and resuspended. Thirty-six microliters of the resuspended beads were added to the libraries and mixed by pipetting. The beads were incubated at room temperature for 5 min and the supernatant was discarded. The beads were washed twice while on the magnet with 200 μL of 80% 200 proof ethanol. After air-drying the beads, the library was eluted off the beads with 23 μL of 0.1× TE and transferred to a new tube. The final library was quantified using a Qubit fluorometer (Applied Biosystems) and analyzed using a Bioanalyzer (Agilent Technologies).

**Chromatin immunoprecipitation sequencing (ChIP-seq).** The antibodies used for ChIP-seq were: H3K27ac (Abcam, Cambridge, MA, USA; Ab4729 lot #GR238071-1), H3K27me3 (Abcam, Cambridge, MA, USA; Ab6002 lot #GR137554-5), H3K4me3 (Abcam, Cambridge, MA, USA; Ab8580 lot #GR240214-1), H3K4me1 (Abcam, Cambridge, MA, USA; Ab8895 lot #GR114265-2), H3K9me3 (Abcam, Cambridge, MA, USA; Ab8898 lot #GR216368-1), ERα (Santa Cruz Biotechnology, Santa Cruz, CA, USA; sc-543X lot #J0313) and CTCF (Cell Signaling Technology, Danvers, MA, USA; D31H2 lot#1). We performed duplicate ChIP-seq experiments for each histone or factor using chromatin collected on different cell culture dates. For each histone ChIP-seq assay, 10 μg of chromatin was incubated with (2.5−5 μg) of antibody. One hundred and fifty micrograms of chromatin was used for CTCF ChIP-seq (with 20 μL of antibody) and 250 μg of chromatin was used for ERα ChIP-seq (with 12 μg of

antibody). ChIP-seq samples were prepared as O'Geen et al. described[59] with minor adjustments. The cells were crosslinked as described above for TCC experiments. The crosslinked cell pellets were washed twice with ice-cold 1× PBS and stored at −80 °C until sonication. Crosslinked cell pellets were thawed on ice and resuspended in 1 mL ice-cold cell lysis buffer (5 mM PIPES pH8, 85 mM KCl, Igepal 10 μL mL⁻¹) containing 1× protease inhibitor cocktail and 1× PMSF. After incubation on ice for 15 min the samples were then homogenized using a 2 mL dounce homogenizer fitted with pestle "B", using 20 strokes. The samples were then centrifuged at 430 rcf for 5 min at 4 °C. The supernatant was removed and the pelleted nuclei were lysed with 1 mL ice-cold nuclei lysis buffer (50 mM Tris-HCl pH8.1, 10 mM EDTA, 1% SDS) containing protease inhibitors (1× protease inhibitor cocktail and 1× PMSF). The nuclei were lysed while incubating on ice for 30 min. Sonication was performed for 12 min using a Covaris Focused-ultrasonicator (Covaris S220) with a peak power of 140 W, duty factor of 10%, and 200 cycles per burst. The sonicated material was then centrifuged at 20,000 rcf for 15 min at 4 °C and transferred to a new tube. To quantify the chromatin, 20 μL of the sonicated chromatin was added to 80 μL of ChIP elution buffer (50 mM NaHCO₃ and 1% SDS) followed by the addition of 12 μL of 5 M NaCl. The samples were boiled at 97 °C for 15 min and 10 μg of RNAseA was added to the tubes once the sample was cooled to room temperature. The sample was incubated at 37 °C for 10 min to allow for RNA digestion. The reverse-crosslinked chromatin was then purified using the Purelink Quick PCR purification kit and eluted in 20 μL of nuclease free water. After quantification via a nanodrop (Thermo Scientific), the total chromatin yield in our sonicated material was calculated. To visualize the fragment sizes of the sonicated chromatin, we ran 1 μg of purified chromatin on a 1.5% agarose gel. If the chromatin fragments were concentrated around the 300−500 bp range, we continued onto immunoprecipitation. If under-sonicated, additional sonication was performed as needed. Five hundred nanograms of purified chromatin sample was saved as our input samples; these samples were brought to a total volume of 150 μL with ChIP elution buffer and stored at −20 °C. ChIP for each target was carried out using the quantities of chromatin and antibody mentioned above. The chromatin for each target was diluted with five times the volume of ice-cold 1× IP dilution buffer (50 mM Tris pH7.4, 150 mM NaCl, 1% Igepal (v/v), 0.25% Deoxycholic acid, 1 mM EDTA pH 8.0) containing protease inhibitors. The appropriate amount of antibody for each reaction was added and rotated overnight at 4 °C. The antibody/chromatin complexes were captured by the addition of 150 μL of protein A/G beads (Pierce #88803), which were first washed twice with 1× IP dilution buffer, for the transcription factor ChIPs and 15 μL of protein A/G beads for the histone ChIPs. These complexes were rotated at 4 °C for 2 h. Following incubation, the beads were captured using a magnetic rack and washed twice with IP wash buffer #1 (50 mM Tris-HCl pH 7.4, 150 mM NaCl, 1% Igepal (v/v) 0.25% Deoxycholic acid and 1 mM EDTA, pH8). The beads were resuspended in the wash buffer for each wash and the supernatant was removed between each wash. The beads were washed three times with IP wash buffer #2 (100 mM Tris-Cl pH9, 500 mM LiCl, 1% Igepal, and 1% Deoxycholic acid). The beads were transferred to a new tube on the third wash. The complexes were eluted off of the beads by the addition of 75 μL of ChIP elution buffer while vortexing at room temperature for 30 min. The supernatant was transferred to a new tube and the elution step was repeated. The ChIP input samples were thawed and 20 μL of 5 M NaCl was added to the 150 μL of final eluted complexes and to the input samples. Crosslinks were reversed overnight at 65 °C and the ChIPs were purified using the Purelink Quick PCR purification kit and the samples were eluted in 35 μL of elution buffer. We performed qPCR against targets enriched for each of the ChIPs. The ChIPs were diluted 1:5 and the input samples were diluted to 1 ng μL⁻¹. Two microliters of DNA was used for each PCR and 1 ng was used for the input sample. Primers against GAPDH were positive for CTCF and H3K4me3; STX16 for CTCF; GREB1 for H3K4me1; TFF1 for H3K4me1, H3K4me3 and H3K27ac; ZNF180 and ZNF333 for H3K9me3; HOXB2 for H3K27me3; and HES3 for H3K27me3. ZNF333 and ZNF180 were negative targets for CTCF, ER-α, H3K4me1, H3K4me3, H3K27ac, and H3K27me3. TFF1 and SHISA5 were used as negative targets for H3K9me3.

The qPCR primers used are as follows:

GAPDH: 5′-CACCGTCAAGGCTGAGAACG-3′ and 5′-ATACCCAAGGGAGCCACACC-3′

STX16: 5′-CCACTCTAATTCAGCGACCA-3′ and 5′-ACTGGGTCCAGGCACTAGG-3′

GREB1: 5′-CACTTTGAGCAAAAGCCACA-3′ and 5′-GACCCAGTTGCCACACTTTT-3′

TFF1: 5′-ATGGGAGTCTCCTCCAACCT-3′ and 5′-TTCCGGCCATCTCTCACTAT-3′

ZNF180: 5′-TGATGCACAATAAGTCGAGCA-3′ and 5′-TGCAGTCAATGTGGGAAGTC-3′

ZNF333: 5′-TGAAGACACATCTGCGAACC-3′ and 5′-TCGCGCACTCATACAGTTTC-3′

HOXB2: 5′-CCAGGCAGACACATAGGAGT-3′ and 5′-ACTGGGCAGAGAAGAGAACC-3′

HES3: 5′-TGTACCTCCCCAGTAGGTGA-3′ and 5′-CTGTCCCAATCCCCGTAAGT-3′

SHISA5: 5′-CAGGACCAGATCGGTGAGTT-3′ and 5′-GCGTGTTCCTCCTCTGATGT-3′

ChIP-seq libraries were generated using the NEBNext ChIP-seq Library Prep Master Mix Set for Illumina (NEB#E6240L) as per the manufacturer's protocol with size selection for the insert size of 300 bp. Half of adaptor-ligated DNA was saved at 4 °C before PCR enrichment of adaptor-ligated DNA. PCR enrichment was done using ten cycles and cleaned with AMPure XP beads at 0.9× as outlined in the protocol. The final library was eluted off of the beads using 30 μL of 0.1× TE and the quality was analyzed with a bioanalyzer (Agilent Technologies).

**Chromosome conformation capture coupled with qPCR (3C-qPCR)**. 3C-qPCR experiment was conducted as Hagège et al. described[60]. Ten million cells (MCF7 or TamR) were harvested and then fixed with 1% formaldehyde for 10 min at room temperature followed by 0.2 M glycine to quench the reaction. Cells were lysed with 0.2% Igepal CA630 for 1 h on ice, then the pelleted nuclei were solubilized with 0.3% SDS for 1 h at 37 °C and diluted with 2% Triton X-100 for 1 h at 37 °C. The genomic DNA was digested with 400 U HindIII overnight at 37 °C and then the digestion was stopped with 1.6% SDS for 20 min at 65 °C. The digested nuclei were diluted with 1:1 volume of ligation buffer and then ligated with 100U T4 DNA ligase. The ligated DNA was de-crosslinked with 300 μg proteinase K overnight at 65 °C and purified by phenol–chloroform extraction. The 3C template was dissolved in 10 mM Tris-HCl and analyzed with the quantitative PCR.

**RNA sequencing (RNA-seq) analysis**. Total RNA were extracted with ZYMO Research Quick-RNA MiniPrep kit. Ten million MCF7 or MCF7-TamR cells were lysed in RNA Lysis Buffer followed by removing the majority of gDNA with Spin-Away Filter. The mixture of RNA and ethanol was then loaded onto Zymo-Spin IIICG Column. Trace DNA was removed by DNase I on the column followed by washing twice with RNA Wash Buffer. The total RNA was eluted with 50 μL DNase/RNase-Free Water. RNA-seq library was prepared with Illumina TruSeq stranded mRNA kit. Four micrograms of total RNA of either parental MCF7 or MCF7-TamR cells was incubated with RNA purification beads and then washed with beads washing buffer. The mRNA was eluted with elution buffer and then reverse transcribed with Superscript III reverse transcriptase. The first strand cDNA was synthesized with first strand synthesis act D mix and the second strand cDNA was synthesized with second strand marking master mix. After cDNA was synthesized, a single adenylate is added to the 3′ end with A-tailing mix and adapters were ligated with ligation mix. DNA fragments were enriched with PCR master mix and then purified to build the DNA library. The library was sequenced with Illumina HiSeq 2000. The differentially expressed genes were identified with CuffDiff[61]. The 50 bp single end sequencing reads were aligned with Tophat module, and then transcripts were assembled with Cufflinks module. The transcript assemblies were compared to annotation with Cuffcompare module. Two or more transcript assemblies were merged with Cuffmerge module. The differentially expressed genes and transcripts were found with Cuffdiff module.

**Identification of compartment patterns and types**. All TCC data were analyzed with HiCLib python package[10] to identify chromatin compartment A or B. Paired-end reads of TCC data were iteratively aligned to human reference genome (hg19) by bowtie2 [62] with the minimal sequencing length of 20 bp and the length step of 5 bp in the module of hiclib mapping. The following reads are removed from the data set in the hiclib HiCdataset object: begin within the 5 bp range from the restriction enzyme cutting site; the duplicate molecules; the fragment pairs separated by less than two restriction sites within the same chromosome; extremely large restriction fragments (more than 10,000 bp) and extremely small restriction fragments (smaller than 100 bp); both ends of pairs starting at exactly the same positions; the top 0.5% most frequently identified restriction fragments. At this stage the self-circles, dangling ends and PCR duplicate reads were removed and maximum molecule length of 500 bp was specified at the initiation of the object. The correlation of two replicates was computed as the following: the counts of mapped reads pairs were accumulated at the 1 M bin, and then the correlation of these counts for each chromosome was calculated separately. After filtering the reads, the frequency contact matrices were constructed at a bin size of 100 Kb with the hiclib fragmentHiC module. The contacts between loci located within the same bin were then removed from the raw heatmap. The bins with less than half of a bin sequenced, the 1% of regions with low coverage were also removed. The top 0.05% of inter-chromosomal counts as the possible PCR blowouts were truncated followed by the performing of iterative correction to get the ICE heatmap with the hiclib binnedData module. All bins of ICE heatmap on a diagonal were removed with the hiclib binnedData module. The bins with less than half of a bin sequenced were also removed. All cis contacts were set to zero to get the trans contacts only. The cis contacts were faked in an interactive way. After removing the bins with zero counts the eigenvector expansion was performed with the hiclib binnedData module to get the first eigenvectors of compartments. The continuous genomic regions of positive first eigenvectors were defined as compartment A (active chromatin), and the continuous genomic regions of negative first eigenvectors were defined as compartment B (inactive chromatin) individually at the 100 K scale. The compartments of five time points (T0, T1, T4, T16, T24) were compared to identify the dynamic changed patterns (Supplementary Fig. 14, S15). First two kinds of compartments: T0 vs. T1 Common and T0 vs. T1 Transit were identified by comparing compartments of T0/T1. The Common compartments are the overlapping

compartments and the Transit compartments are differential compartments, which will be used in the following steps as well. Next, T0 vs. T1 Common and T0 vs. T1 transit were compared with T4, T16, T24 independently to generate the (a) T0 vs. T1 Common vs. T4/T16/T24 Common, (b) T0 vs. T1 Common vs. T4/T16/T24 Transit, (c) T0 vs. T1 transit vs. T4/T16/T24 Common, (d) T0 vs. T1 transit vs. T4/T16/T24 Transit. Thirdly, the patterns 1−15 were produced by comparing the various time points (T4, T16 and T24) of last step subsets a, b, c, and d, which are vs. T4, vs. T16, vs. T24. The rest subsets were divided into patterns 16−24 according to the numbers of converted bins. Finally, 24 patterns were identified from the intersection and difference among subsets a, b, c, and d. According to their biological meanings, these patterns were able to categorize into six types of dynamic changed compartments (DCCs): HCC (patterns 1−4), ETC (patterns 5−8), LTC (patterns 9−12), LDC (patterns 13−16), MDC (patterns 17−20), HDC (patterns 21−24).

**Computation of differential compartments**. The variance of first eigenvector values of compartment in T0, T1, T4, T16, T24 was computed at the 100 K scale. The difference of HCC/ETC/LTC/LDC with MDC/HDC was determined by two-sided Wilcoxon rank-sum test for their averaged variance of first eigenvector values. The estimate of FDR of the differential compartments between two compartments was conducted by a permutation-based test. In brief, the difference in means of eigenvector values of each of two compartments: Compartment 1 and Compartment 2, was first calculated as the observed value. The eigenvector values of two compartments were then pooled together and randomly selected one half as randomized compartment 1 and the other half as randomized compartment 2. This was done for a total of 1000 rounds of permutation. In each round, the difference in means of eigenvector values of each of two randomized compartments was calculated as a permuted value. All permutated values were combined into a null distribution. The FDR is estimated based on how many permutated values are above the observed value and the permutated null.

**Epigenetic states**. ChIP-seq of H3K4me3, H3K27ac, H3K4me1, H3K27me3, and H3K9me3 data sets in five time points E2-induced MCF7 cells and TamR cells were aligned to human hg19 genome and then trained by a Java program ChromHMM v1.17 [45]. ChromHMM is able to learn and characterize chromatin epigenetic states by integrating multiple ChIP-seq data sets of various histone modification marks to identify de novo the major re-occurring combinatorial and spatial patterns based on a multivariate hidden Markov model. The results of the model can then be used to systematically annotate genome-wide maps of chromatin state which facilitate the biological meaning in one or more cell types. After ChIP-seq data were mapped with HG19 human genome, the BinarizeBam module was used to binarize uniquely mapped reads into 1 K length bins for model learning. The binarized data were then trained with LearnModel module and ten epigenetic states were finally identified with a minimum $p$ value after averaging five times training. The emission and transition matrices were visualized by the R program. The ten ChromHMM states were classified into three kinds of epigenetic states according to the combination of histone marks with the $p$ value cutoff of emission matrices at 0.3: active states defined by H3K4me3, H3K27ac and H3K4me1, repressive states defined by H3K27me3, H3K9me3, and bivalent states including both active and repressive states.

**Identification of ERα and CTCF binding sites (peaks)**. ChIP-seq data of ERα and CTCF were aligned to human hg19 genome followed by peaks calling with MACS v1.4.2[46]. The identified peaks were located to the various compartments or their boundary. The peak summits generated by MACS were defined as binding sites for the subsequent analysis.

**Differential binding analysis of ERα**. ERα differential peaks (DPs) were identified with R Bioconductor package DiffBind v2.6.6[63,64] using TamR vs. T0/T1/T4/T16/T24 as the contrast. There are the following steps for DiffBind processing differential binding analysis. Peak callers-identified peaks (specific protein binding sites) were firstly enriched for genomic loci from ChIP-seq data and then been read by DiffBind. Secondly, overlaps of peaks were examined to determine how well similar samples cluster together with the function dba.count. Thirdly, overlap reads in each interval for each unique sample were counted with the function dba.contrast. Fourthly, a contrast (or contrasts) is established and then the core analysis of DiffBind was executed by default using DESeq2[65] with the function dba.analyze. Finally, the results were reported and plotted with the function dba.report.

**Time series RNA-seq data analysis**. Time series RNA-seq data of E2-treated MCF7 cells[47] were acquired from GSE62789. Five time points close to this study (T0′ = 0 min, T1′ = 40 min, T4′ = 160 min, T16′ = 640 min, T24′ = 1280 min) were selected and mapped with Tophat and gene expression was analyzed with Cuffdiff[61]. After gene expression values were normalized with log2, then the variances were calculated from five time points. The normalized gene expression values of genes located at each compartment types in five time points were visualized with the R program.

**Significant interaction loops**. The uniquely mapped paired-end reads were input to HOMER v4.7[49] to generate the significant interaction loops. In brief, HOMER was originally developed for a de novo motif discovery program and now was able to identify significant loops. HOMER analyzeHiC module was used to make interaction matrices and normalize interaction counts, then identify the significant interaction loops. The loops were further filtered with LogP cutoff -6 and distance cutoff 20 Kb between loci centers. Tamoxifen-resistant differential loops of MCF7-TamR were obtained by analyzing HiC module of HOMER using MCF7 T0 as the contrast with the cutoff of FDR ≤ 0.1 and the distance of loci pair center at 40 K to 5 M.

**Enrichment of KEGG pathway**. Differentially expressed genes in various compartments were enriched by gene set enrichment analysis (GSEA) v3.0[48]. Kyoto Encyclopedia of Genes and Genomes (KEGG) were selected as gene sets database. Gene ranking was determined by the ratio of log2 fold change to $p$ value of differential expression.

**Relapse-free survival analysis**. Kaplan−Meier plot of genes was generated by an online tool and resource[66] on the website (http://www.kmplot.com) with the probability of relapse-free survival in ER+ patients receiving only Tamoxifen but without chemotherapy, who were stratified by mRNA levels at the top quartile (25%) vs. the rest (75%). $p$ value was computed with the log-rank test.

**Reporting summary**. Further information on experimental design is available in the Nature Research Reporting Summary linked to this article.

## Data availability

Raw and processed TCC, ChIP-seq and RNA-seq data for MCF7 and TamR cells are deposited in GEO under accession number GSE108787, and raw and processed TCC data for T47D and TamR cells are deposited in GEO under accession number GSE119890. The RNA-seq data of E2-treatment time series MCF7 are available at GEO accession number GSE62789.

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

## Acknowledgements

We thank the University of Vermont and UTHSA Next Generation Sequencing Facilities for services rendered for production of the TCC, ChIP-seq and RNA-seq data. This project was partially supported by grants from NIH R01GM114142 (V.X.J.), U54CA217297 (V.X.J., S.F.), R01GM129338 (S.F.), ACS Institutional Research Grant 14-196-01 (S.F.); and NIH P50CA186784 (SPORE) (R.S.), Breast Cancer Research Foundation (BCRF) 16-142, 17-143, 18-145 (R.S.), DOD W81XWH-14-1-0326 (X.F.).

## Author contributions

V.X.J. and S.F. conceived the project. D.L.G., S.F., A.J.F., Y.Z., T.L. and Y.Y. conducted the experiments. Y.Z. performed the data analysis. V.X.J., Y.Z. and S.F. wrote the manuscript, with all authors including J.W., M.R., X.F., R.S. and S.L. contributing to writing and providing the feedback.

## Additional information

**Competing interests:** Rachel Schiff reported research funding from AstraZeneca, GlaxoSmithKline, Gilead Sciences, and PUMA Biotechnology, and is a consulting/advisory committee member for Macrogenics and Eli Lilly outside the submitted work. The remaining authors declare no competing interests.

