## [Peer Review File · Nature Communications]

Reviewers' comments:

Reviewer #1 (Remarks to the Author):

The authors develop a deep resource to probe 3D architecture of transcription factor activity in E2 sensitive and resistant lines. The resource would be of interest to the community. The overall conclusions from the manuscript are not particularly novel or innovative. Rather the manuscript should be seen as a resource for the community.

Specific comments

1. The abstract as written is essentially unintelligible to the average reader in the absence of having read the manuscript or being an expert in the narrow field of study presented. It needs to be rewritten to be clear to the average reader. Cryptic comments such as "compartment A" and "decreased CTCF insulation activity" would be confusing to all but experts in the field and do not contain sufficient information to explain the concepts.

2. The authors perform studies exclusively in MCF7 cells. This is a tractable system and has been used widely. However, this cell line represents a single "incident" and the data should be extended to other cells and preferably to a primary cell system to establish generalizability.

3. The authors need to define criteria that would result in allocation to different compartments ie

Unexpectedly, we were able to categorize these 24 patterns into six types of biological meaningful temporal dynamic re- compartmentalization (TDRC): Highly Common Compartments (HCC, patterns 1-4), Early Transit Compartments (ETC, patterns 5-8), Late Transit Compartments (LTC, patterns 9-12), Lowly Dynamic Compartments (LDC, patterns 13-16), Moderately Dynamic Compartments (MDC, patterns 17-20), and Highly Dynamic Compartments (HDC, patterns 21-24) (Figure 2C).

What is the evidence that these are "biologically meaningful". The term biologically relevant and biologically meaningful is used at multiple points in the manuscript. What is the data that supports this contention.

4. The authors describe changes between with and without E2 and in E2 sensitive and resistant cells. They do not provide any evidence that these changes are statistically significant. Given the number of analyses, the observational nature of the studies are concerning. Indeed, there are only 2 measurements at each time point and a single cell line. Are the proposed changes in 3D architecture generalizable and robust.

5. The authors claim an association with particular biological functions of genes located in dynamic regions. Are these associations significant in that if a random set of genes was chosen (the n in this set is large, would there be a similar set of associations with particular cellular processes).

6. The authors state "Our data implied that these moderate to higher dynamically changed compartments may play an essential role in governing this hormone-mediated luminal breast cancer development. " What is the data to support this contention. There is nothing in this paper related to breast cancer development.

7. The authors state: Interestingly, two types of temporal dynamic re-compartmentalization (TDRC), i.e., moderately and highly dynamic compartments (MDCs and HDCs), showed higher alteration in TamR cells (Figure 3B). I agree this is interesting. However, why would there be greater re compartmentalization in TamR cells. What is the meaning of the observation.

8. The authors state: Although our result is not so surprising, it nevertheless implicates an underlying molecular event that ER α regulated highly re-compartmentalization may require to loosen CTCF insulator at the domain boundary (Zuin et al. 2014; Nora et al. 2017). What is the chicken and the egg. Is it re-compartmentalization that alters CTCF or vice versa. Indeed, I am not convinced by the data as presented that the CTCF insulator is responsible or markedly altered in terms of re compartmentalization.

In summary this observational paper provides a strong database for future studies. The authors actually suggest a number of studies that would convert this from an observational to a mechanistic paper. The conclusions and models presented are speculative and solely based on observational data. The conclusions drawn such as relationships to biological relevance and disease initiation or progression is not supported by the data as presented.

Reviewer #2 (Remarks to the Author):

Zhou et al characterize 3D chromatin dynamics during E2-induction of MCF7 and TamR cell lines in the paper "Temporal dynamic reorganization of 3D chromatin architecture in hormone-induced breast cancer and endocrine resistance". They provide many new low resolution tethered chromatin capture (~Hi-C) data sets as well as a large number of new CHIP-seq data sets targeting various histone marks. Their main findings pertain to classes of genomic regions that undergo compartment

switching and/or shifting of boundaries delineating compartmentalization. This manuscript has one major crack in its foundation that I believe it is important to address. While the results are intriguing, virtual all of their findings stem from the characterization of differential compartmentalization. The classification of differential compartmentalization was not, as far as I can tell, rooted in any statistical method and did not consider variability between data sets. They also did not provide any measure of false discovery. Since the whole paper relies on these differential regions, these issues should be addressed before publication.

Major Comments:

1) The methods to determine differential compartmentalization are not based on any statistical methods, are not controlled for false positives, and are therefore likely to contain many (perhaps mostly) false positives. The authors classified every region of the genome as either belonging to the A or B compartment based on principle component analysis which is fine; however; their method to determine changes in these regions appeared to be simply looking for any change in boundaries (I am guessing even a shift of a single bin) or reclassification from one compartment to another. Granted, it is hard to know exactly how they determined these differences from the text in their methods section (see below).

“The compartments of five time points (T0, T1, T4, T16, T24) were compared to identify the dynamic changed patterns. First, the T0/T1 Common and T0/T1 Transit compartments were obtained from comparing T0 and T1; Then, T0/T1/T4, or T0/T1/T16 or T0/T1/T24 Common and Transit compartments were further derived from comparing T0/T1 Common/Transit compartments with T4, T16 and T24 respectively. Finally, 24 patterns were identified from the intersection and difference among these subsets. According to their biological meanings, these patterns were able to categorize into six types of dynamic changed compartments (DCCs): HCC (patterns 1-4), ETC (patterns 5-8), LTC (patterns 9-12), LDC (patterns 13-16), MDC (patterns 17-20), HDC (patterns 21-24).”

Regions that were classified as A but were only slightly above the middle line in their compartmentalization plots (like the one shown at the very top of Figure 1A) could easily be classified as compartment B in a different sample. But that change may not be statistically significant. This could also be true for slight shifting of boundaries. Moreover, you would expect more of this shifting and differential classification as noise increases (i.e. as you use smaller and smaller bins) which is exactly what the authors observe at the bottom of page 5.

The remedy for this situation is to use a statistical method that assesses variability between biological replicates and uses that to determine statistically significant changes in compartmentalization between data sets. There may be a tool that does this but I am not aware of

it. The authors could devise a reasonable method themselves. It need not be perfect but it should be reasonable. A good starting point would be to apply the same method they are currently using to determine “Transit” compartments to identify Transit compartments between biological replicates. These would obviously be false positives. Then they should compare two samples that are not biological replicates (important, only use one replicate of each sample). Do this for all pairs of replicates and all pairs of samples. Use the (median # between-replicate differences) / (median # between-sample differences) as a decent starting point for false discovery rate. This is a very rough estimate that statisticians would likely cringe at. But is an easy value to get and would be a good starting point. But I would strongly encourage the recruitment of a biostatistician to devise a more robust and accurate method. Since the whole paper relies on these differences it is absolutely essential to get reasonably accurate differential compartment assignments with a reasonable estimate of false discovery rate.

2) The authors describe way more loops than are thought to exist in a single cell type using data that is lower resolution than typically required for loop detection. Rao et al (Cell 2014) elegantly showed how APA plots can be used to determine loop calling accuracy. In a supplemental figure from that paper (figure S7 I think) they showed that most loop calling approaches were mostly false positives. In situ Hi-C sequenced to one billion reads or more has detected only about 10,000 loops per cell type. Most of these loops are < 2 Mb in length. The authors of this paper claim to have identified 62,000 loops from data sequenced to much lower depth. While I could not find more information about the loops I did look at Table S4 which listed 420 loops. These loops averaged 138 Mb in length, far larger than loops detected by deeply sequenced in situ Hi-C. These loops seem questionable at best.

Minor Comments:

1) I understand how the authors classified the genome into two compartments. But it is not clear from the methods how they assigned each of the two compartments to A and B.

2) The terminology used regarding compartments throughout the paper is confusing. My understanding is that there are two compartments and that regions of the genome can belong to one or the other. But the authors describe the identification of ~2000 compartments. I think it is actually 2000 distinct genomic regions that were assigned to one of two compartments. I am not confident of the best way to describe this. But my impression is that 2000 compartments is incorrect and confusing.

3) The figures were not labeled (e.g. Figure 1, Figure 2, etc). I had to know which figure was which based on the order. Also, I would greatly prefer if the file was single spaced and had embedded

figures with captions below them. It is very hard to read the paper in its current format. I realize that this may be the guidelines provided by the journal. But I would encourage authors to disregard those guidelines. Your reviewers will appreciate it.

4) There was no reference to table S4 but there were references to file S4. There were also references to Table S5 which I don't think exists. Admittedly I may have missed something but it is worth double checking.

5) I could not find a description of what each table was (i.e. caption).

6) The authors used a lot of acronyms which made the paper very hard to follow. In general I felt that there were too many subsets of data. We would recommend that the Results be distilled more rather than plotting every possible comparison of data sets.

7) Correlation is not a great way to determine reproducibility or quality of Hi-C data sets. It is dominated by distance dependent contact frequencies. It is ok to show. And slightly reassuring that the experiments worked but not great for determining data quality.

8) ChIP seq comparisons seemed to be based on raw overlap of peaks. Comparisons between data sets require statistical methods to call differential peaks (e.g. EdgeR or DESeq2).

9) The methods section was quite brief. I tried to look up how correlations were calculated for figure 1 but found no description. The methods describing detection of Transit compartments was very brief and unclear. There was no description of how differential ChIP seq analysis was done. Given the nature of this paper I would expect the methods section to be many times longer than it is.

Response to Reviewers

Reviewer #1 (Remarks to the Author):

1. The abstract as written is essentially unintelligible to the average reader in the absence of having read the manuscript or being an expert in the narrow field of study presented. It needs to be rewritten to be clear to the average reader. Cryptic comments such as "compartment A" and "decreased CTCF insulation activity" would be confusing to all but experts in the field and do not contain sufficient information to explain the concepts.

Response: We have revised the abstract and used terminologies for general readers.

2. The authors perform studies exclusively in MCF7 cells. This is a tractable system and has been used widely. However, this cell line represents a single "incident" and the data should be extended to other cells and preferably to a primary cell system to establish generalizability.

Response: We have performed the TCC in T47D cell line upon the stimulation of E2 at 0 hour, 1 hour, 4 hour, 16 hour and 24 hour as well (unpublished data). We were able to identify the same 24 patterns of temporal dynamic 3D chromatin conformation in T47D cells as in MCF7 cells (see the attached **Figure R1**). This suggests our studies are generalizable in various cell lines. However, there are slightly different numbers of compartments A or B in each of 24 patterns between two cancer cell lines. The detailed comparison is beyond of the scope of this work and the results and biological meaningful interpretations will be presented in the next manuscript.

Figure R1. A total of 24 temporal dynamic 3D chromatin patterns were identified in T47D cell line.

3. *The authors need to define criteria that would result in allocation to different compartments. Unexpectedly, we were able to categorize these 24 patterns into six types of biological meaningful temporal dynamic re- compartmentalization (TDRC): Highly Common Compartments (HCC, patterns 1-4), Early Transit Compartments (ETC, patterns 5-8), Late Transit Compartments (LTC, patterns 9-12), Lowly Dynamic Compartments (LDC, patterns 13-16), Moderately Dynamic Compartments (MDC, patterns 17-20), and Highly Dynamic Compartments (HDC, patterns 21-24) (Figure 2C). What is the evidence that these are "biologically meaningful". The term biologically relevant and biologically meaningful is used at multiple points in the manuscript. What is the data that supports this contention.*

Response: Many studies have shown that the dysregulated functions of Estrogen Receptor α (ER α) signaling lead to the ER α + endocrine resistant breast cancer (Osborne et al., 2001. Nemere et al. 2003., Levin et al., 2015. Lupien et al., 2010. Ross-Innes et al., 2012. Fu et al., 2016). This may due to the altered crosstalk between ER and growth factor receptor kinase pathway, the alteration of ER co-regulators, and the ER co-regulators related transcriptional reprogramming. However, no studies so far have been focused on the roles of 3D chromatin in endocrine resistant breast cancer. Our work has filled up this gap and addressed this essential biological meaningful question: our findings have indicated that ER α transcriptional activity is increased but CTCF insulation activity is decreased in higher dynamic chromatin conformation (**Fig. 5**). The

enriched signaling pathways of the genes in higher dynamic chromatin conformation may have a strong effect on endocrine resistance (**Fig. 6, Suppl. Figs. S21-25**). The higher expression of the genes in higher dynamic chromatin conformation leads to the lower survival of tamoxifen resistant patients (**Suppl. Figs. S26-28**). All these data clearly supported the contention that our results are biological meaningful.

References:

Osborne CK, Schiff R, Fuqua SA, Shou J. Estrogen receptor: current understanding of its activation and modulation. *Clin cancer Res: Official J Am Assoc Cancer Res* 2001;7:4338se42s.

Nemere I, Pietras RJ, Blackmore PF. Membrane receptors for steroid hormones: signal transduction and physiological significance. *J Cell Biochem* 2003;88:438e45.

Levin ER. Extranuclear steroid receptors are essential for steroid hormone actions. *Annu Rev Med* 2015;66:271e80.

Lupien M, Meyer CA, Bailey ST, Eeckhoute J, Cook J, Westerling T, et al. Growth factor stimulation induces a distinct ER(alpha) cistrome underlying breast cancer endocrine resistance. *Genes & Dev* 2010;24:2219e27.

Ross-Innes CS, Stark R, Teschendorff AE, Holmes KA, Ali HR, Dunning MJ, et al. Differential oestrogen receptor binding is associated with clinical outcome in breast cancer. *Nature* 2012;481:389e93.

Fu X, Jeselsohn R, Pereira R, Hollingsworth EF, Creighton CJ, Li F, Shea M, Nardone A, De Angelis C, Heiser LM, Anur P, Wang N, Grasso CS, Spellman PT, Griffith OL, Tsimelzon A, Gutierrez C, Huang S, Edwards DP, Trivedi MV, Rimawi MF, Lopez-Terrada D, Hilsenbeck SG, Gray JW, Brown M, Osborne CK, Schiff R. FOXA1 overexpression mediates endocrine resistance by altering the ER transcriptome and IL-8 expression in ER-positive breast cancer. *Proc Natl Acad Sci U S A*. 2016 Oct 25;113(43):E6600-E6609.

4. The authors describe changes between with and without E2 and in E2 sensitive and resistant cells. They do not provide any evidence that these changes are statistically significant. Given the number of analyses, the observational nature of the studies are concerning. Indeed, there are only 2 measurements at each time point and a single cell line. Are the proposed changes in 3D architecture generalizable and robust.

Response: We now added a Wilcoxon rank-sum test for the variance of first eigenvectors of the 3D architecture between lower dynamic regions and higher dynamic

regions in both E2 stimulated cells (**Fig. 2C**) and tamoxifen resistant cells (**Suppl. Fig. S16**). Both results showed that the changes of 3D genome architecture are statistically significant.

5. The authors claim an association with particular biological functions of genes located in dynamic regions. Are these associations significant in that if a random set of genes was chosen (the n in this set is large, would there be a similar set of associations with particular cellular processes).

Response: We have performed GSEA (Gene Set Enrichment Analysis) on the random set of genes (n = 2000) in dynamic compartments for three times, and none of interesting signaling pathways was significantly enriched.

6. The authors state "Our data implied that these moderate to higher dynamically changed compartments may play an essential role in governing this hormone-mediated luminal breast cancer development. " What is the data to support this contention. The is nothing in this paper related to breast cancer development.

Response: To ease this concern, we additionally performed the survival analysis of three genes (SOX4, GPRC5C, PRC1) in higher dynamically changed compartments (**Suppl. Figs. S26-S28**). Analysis was referred to the published paper (Lanczky et al. 2016). Our results showed that there are statistically significant ($P \leq 0.05$, log-rank test) for the relapse-free survival probability between the hormone only breast cancer patients at the top 25% mRNA level of genes and hormone only breast cancer patients at the low 75% mRNA level of genes. SOX4 has been reported to induce Epithelial–Mesenchymal Transition and contributes to cancer progression in breast cancer cells (Zhang et al., 2012, Tiwari et al., 2013). Both GPRC5C and PRC1 have been shown to be involved in the growth of breast cancer (Yamaga et al., 2014. Shimo et al., 2007). Thus, our data implied that genes in higher dynamic chromatin may play an essential role in the hormone-mediated luminal breast cancer development.

References:

Lanczky A, Nagy A, Bottai G, Munkacsy G, Paladini L, Szabo A, Santarpia L, Gyorffy B. (2016) miRpower: a web-tool to validate survival-associated miRNAs utilizing expression data from 2,178 breast cancer patients. *Breast Cancer Res Treat.* 160(3):439-446.

Zhang J, Liang Q, Lei Y, Yao M, Li L, Gao X, Feng J, Zhang Y, Gao H, Liu DX, Lu J, Huang B. SOX4 induces epithelial-mesenchymal transition and contributes to breast cancer progression. *Cancer Res.* 2012 Sep 1;72(17):4597-608.

Tiwari N, Tiwari VK, Waldmeier L, Balwierz PJ, Arnold P, Pachkov M, Meyer-Schaller N, Schübeler D, van Nimwegen E, Christofori G. Sox4 is a master regulator of epithelial-mesenchymal transition by controlling Ezh2 expression and epigenetic reprogramming. *Cancer Cell.* 2013 Jun 10;23(6):768-83.

Yamaga R, Ikeda K, Boele J, Horie-Inoue K, Takayama K, Urano T, Kaida K, Carninci P, Kawai J, Hayashizaki Y, Ouchi Y, de Hoon M, Inoue S. Systemic identification of estrogen-regulated genes in breast cancer cells through cap analysis of gene expression mapping. *Biochem Biophys Res Commun.* 2014 May 9;447(3):531-6.

Shimo A, Nishidate T, Ohta T, Fukuda M, Nakamura Y, Katagiri T. Elevated expression of protein regulator of cytokinesis 1, involved in the growth of breast cancer cells. *Cancer Sci.* 2007 Feb;98(2):174-81.

7. The authors state: Interestingly, two types of temporal dynamic re-compartmentalization (TDRC), i.e., moderately and highly dynamic compartments (MDCs and HDCs), showed higher alteration in TamR cells (Figure 3B). I agree this is interesting. However, why would there be greater re compartmentalization in TamR cells. What is the meaning of the observation.

Response: Our observation may indicate that in TamR cells, the ER co-regulators or the crosstalk between ER and growth factor receptor kinase pathway may lead to the re-compartmentalization of genome; Or the re-compartmentalization of genome may alter the downstream gene expression of ER co-regulators which lead to tamoxifen resistance. In either case, the re-compartmentalization is essential. Illustration and identification of the re-compartmentalization are very helpful for the further mechanistically studying the tamoxifen resistance and for developing the new therapeutic strategy.

8. The authors state: Although our result is not so surprising, it nevertheless implicates an underlying molecular event that ER α regulated highly re-compartmentalization may require to loosen CTCF insulator at the domain boundary (Zuin et al. 2014; Nora et al. 2017). What is the chicken and the egg. Is it re-compartmentalization that alters CTCF or vice versa. Indeed, I am not convinced by the data as presented that the CTCF insulator is responsible or markedly altered in terms of re compartmentalization.

Response: How does CTCF insulator play the role for the 3D genome architecture? This is a very interesting topic in the genome research field and many investigators focus on the mechanism of CTCF and the conclusion is open and controvertible so far. For example, Tang et al. assumed that CTCF mediates the 3D genome architecture, Nora et al. found the degradation of CTCF decouples the genomic loops, Gong et al. suggested that higher genomic boundary insulation scores are associated with elevated CTCF levels and that they may differ across cell types.

References:

Tang Z, Luo OJ, Li X, Zheng M, Zhu JJ, Szalaj P, Trzaskoma P, Magalska A, Wlodarczyk J, Ruszczycki B, Michalski P, Piecuch E, Wang P, Wang D, Tian SZ, Penrad-Mobayed M, Sachs LM, Ruan X, Wei CL, Liu ET, Wilczynski GM, Plewczynski D, Li G, Ruan Y. CTCF-Mediated Human 3D Genome Architecture Reveals Chromatin Topology for Transcription. *Cell*. 2015 Dec 17;163(7):1611-27.

Nora EP, Goloborodko A, Valton AL, Gibcus JH, Uebersohn A, Abdennur N, Dekker J, Mirny LA, Bruneau BG. Targeted Degradation of CTCF Decouples Local Insulation of Chromosome Domains from Genomic Compartmentalization. *Cell*. 2017 May 18;169(5):930-944.e22.

Gong Y, Lazaris C, Sakellaropoulos T, Lozano A, Kambadur P, Ntziachristos P, Aifantis I, Tsirigos A. Stratification of TAD boundaries reveals preferential insulation of super-enhancers by strong boundaries. *Nat Commun*. 2018 Feb 7;9(1):542. doi: 10.1038/s41467-018-03017-1.

Reviewer #2 (Remarks to the Author):

Major Comments:

1) *The methods to determine differential compartmentalization are not based on any statistical methods, are not controlled for false positives,*

“The compartments of five time points (T0, T1, T4, T16, T24) were compared to identify the dynamic changed patterns.....”

Regions that were classified as A but were only slightly above the middle line in their compartmentalization plots (like the one shown at the very top of Figure 1A) could easily be classified as compartment B in a different sample.....

The remedy for this situation is to use a statistical method that assesses variability between biological replicates and uses that to determine statistically significant changes in compartmentalization between data sets....

Response: We now conducted the statistical analysis for the compartments. In E2 induced MCF7 cells, the combined MDC and HDC have a higher variance of first eigenvectors than other compartment types, and Wilcoxon rank-sum test suggests there is a statistically significant difference ($p=7.6 \times 10^{-11}$) between higher dynamic compartments and lower dynamic compartments (**Fig. 2C**). And when tamoxifen resistant MCF7 cells are also included, higher dynamic compartments have greater variance of first eigenvectors than that of lower dynamic compartments ($p=9.3 \times 10^{-6}$, Wilcoxon rank-sum test, **Suppl. Fig. S16**).

2) The authors describe way more loops than are thought to exist in a single cell type using data that is lower resolution than typically required for loop detection. Rao et al (Cell 2014) elegantly showed how APA plots can be used to determine loop calling accuracy. In a supplemental figure from that paper (figure S7 I think) they showed that most loop calling approaches were mostly false positives. In situ Hi-C sequenced to one billion reads or more has detected only about 10,000 loops per cell type. Most of these loops are < 2 Mb in length. The authors of this paper claim to have identified 62,000 loops from data sequenced to much lower depth. While I could not find more information about the loops I did look at Table S4 which listed 420 loops. These loops averaged 138 Mb in length, far larger than loops detected by deeply sequenced in situ Hi-C. These loops seem questionable at best.

Response: Normally the putative promoter-enhancer interactions in a typical cell type are far more than 100,000. For example, Ron et al. predicted 179,745 putative enhancers regulated a total of 20,264 genes in human. In our case, 42,390 putative differential loops of TamR were obtained by analyzing HiC module of HOMER with the cutoff of $FDR \leq 0.1$ and the distance of loci pair center at 40K to 5M. As we described in **Fig. 6E**, there are 16,807 loops with gene promoters. Of them, 9,638 loops with promoter-enhancer which are assumed as “active” loops, a number similar to the “active” 10,000 loops per cell type identified by Rao et al. Furthermore, we have identified 4,122 loops with ER α -related promoter-enhancer, and 599 loops with ER α -related promoter-enhancer and 396 associated differentially expressed genes located at TA-MDC&TA-HDC.

Reference:

Ron G, Globerson Y, Moran D, Kaplan T. Promoter-enhancer interactions identified from Hi-C data using probabilistic models and hierarchical topological domains. Nat Commun. 2017 Dec 21;8(1):2237.

Minor Comments:

1) I understand how the authors classified the genome into two compartments. But it is not clear from the methods how they assigned each of the two compartments to A and B.

Response: We identified the compartments with HiClib Python package (Imakaev et al., 2012). Briefly, after the paired-end raw reads were iteratively mapped to hg19 genome with bowtie2, self-circles, dangling ends and PCR duplicate reads were filtered to remove and then iterative correction and eigenvector decomposition were performed. The continuous genomic regions of positive first eigenvector were defined as compartment A (open or active chromatin) and the continuous genomic regions of negative first eigenvector was defined as compartment B (close or repressed chromatin) at the 100K scale.

Reference:

Imakaev, M., Fudenberg, G., McCord, R.P., Naumova, N., Goloborodko, A., Lajoie, B.R., Dekker, J., and Mirny, L.A. (2012). Iterative correction of Hi-C data reveals hallmarks of chromosome organization. Nat. Methods 9, 999–1003.

2) The terminology used regarding compartments throughout the paper is confusing. My understanding is that there are two compartments and that regions of the genome can belong to one or the other. But the authors describe the identification of ~2000 compartments. I think it is actually 2000 distinct genomic regions that were assigned to one of two compartments. I am not confident of the best way to describe this. But my impression is that 2000 compartments is incorrect and confusing.

Response: We now updated the description of compartments though the definition was not changed. To make it clearer, we defined the continuous genomic regions of positive first eigenvector (PCA analysis in HiClib Python package) as compartment A (open chromatin), the continuous compartments of negative first eigenvector (PCA analysis in HiClib Python package) as compartment B (close chromatin). In a genome-wide scale, ~2,000 compartments are either compartment A or compartment B. Each certain genomic region is only assigned to one compartment (A or B).

3) The figures were not labeled (e.g. Figure 1, Figure 2, etc). I had to know which figure was which based on the order. Also, I would greatly prefer is the file was single spaced

and had embedded figures with captions below them. It is very hard to read the paper in its current format. I realize that this may be the guidelines provided by the journal. But I would encourage authors to disregard those guidelines. Your reviewers will appreciate it.

Response: It is a good idea for a manuscript to demonstrate the figures as the reviewer commented so that the reviewers could read them easier and more comfortable.

4) There was no reference to table S4 but there were references to file S4. There were also references to Table S5 which I don't think exists. Admittedly I may have missed something but it is worth double checking.

Response: We updated all the Suppl. Tables, and Files. The Suppl. Tables are in the Suppl. Information and Suppl. Files are the Excel file for the lists.

5) I could not find a description of what each table was (i.e. caption).

Response: We added the description for each of five suppl. tables.

6) The authors used a lot acronyms which made the paper very hard to follow. In general i felt that there were too many subsets of data. We would recommend that the Results be distilled more rather than plotting every possible comparison of data sets.

Response: We agreed that we have generated a huge amount of data, including 1.3 billion 3D genome data, more than 3.5 billion ChIP-seq of histone modification, ER and CTCF data and more than 210 million RNA-seq data. Indeed, it's not easy to organize them perfectly. All these results are essential to our paper. We did our best to present the results in details to make the reader easier to understand the 3D genomic conformation in endocrine resistant breast cancer.

7) Correlation is not a great way to determine reproducibility or quality of Hi-C data sets. It is dominated by distance dependent contact frequencies. It is ok to show. And slightly reassuring that the experiments worked but not great for determining data quality.

Response: We performed the Negative Binomial dispersion of T0/T1 two replicates as the supplement (**Suppl. Fig. S1**). After TCC raw data were mapped to human hg19 genome, read pairs of two replicates of T0 and T1 were treated with diffHiC (Lun ATL and Smyth GK, 2015) to estimate the Negative Binomial dispersion for the modelling

biological variability. The variation is obviously decreased when the counts are increased due to better precision.

Reference:

Lun ATL, Smyth GK (2015). diffHic: a Bioconductor package to detect differential genomic interactions in Hi-C data. BMC Bioinformatics, 16, 258.

8) ChIP seq comparisons seemed to be based raw overlap of peaks. Comparisons between data sets require statistical methods to call differential peaks (e.g. EdgeR or DESeq2).

Response: We performed the differential binding analysis of ER α in tamoxifen resistant altered compartments (**Suppl. Fig. S17**). ER α differential peaks (DPs) were identified with DESeq2 based DiffBind (Stark and Brown, 2011. Ross-Innes CS et al, 2012.) using TamR vs. T0/T1/T4/T16/T24 as the contrast. The numbers of DPs in each temporal dynamic re-compartmentalization (TDRC) was calculated. Among them the percentage of DPs within tamoxifen resistant altered compartment (TRAC) was demonstrated. The results showed that there were more ER α differential peaks in higher dynamic chromatin.

References:

Stark R, Brown G (2011). DiffBind: differential binding analysis of ChIP-Seq peak data. <http://bioconductor.org/packages/release/bioc/vignettes/DiffBind/inst/doc/DiffBind.pdf>.

Ross-Innes CS, Stark R, Teschendorff AE, Holmes KA, Ali HR, Dunning MJ, Brown GD, Gojis O, Ellis IO, Green AR, Ali S, Chin S, Palmieri C, Caldas C, Carroll JS (2012). Differential oestrogen receptor binding is associated with clinical outcome in breast cancer. Nature, 481, -4.

9) The methods section was quite brief. I tried to look up how correlations were calculated for figure 1 but found no description. The methods describing detection of Transit compartments was very brief and unclear. There was no description of how differential ChIP seq analysis was done. Given the nature of this paper I would expect the methods section to be many times longer than it is.

Response: We updated the methods section in much more details.

Reviewers' comments:

Reviewer #1 (Remarks to the Author):

I am not convinced that the authors have adequately resolved the concerns of the reviews.

The abstract is modestly improved but still requires editing for English syntax and clarity.

The authors were asked to demonstrate generalizability of the data beyond the MCF7 cell line. Providing data in the rebuttal and not including this in the manuscript is not acceptable in today's climate where generalizability is critical. Indeed, there are marked differences between the results in T47D and MCF7 that are important to characterize and understand and importantly to determine which represent a generalizable event. Thus indicating that this information is beyond the scope of this work.

Both reviewers asked the authors to describe and develop clear and statistically robust methods to explain how the 24 patterns were defined and then how they were compartmentalized into six types of biologically meaningful compartments. The response of the authors did not answer the question. How were the patterns defined and how were they compartmentalized and further how what "biologically meaningful" defined.

The authors were asked to provide a reason why there would be greater re-compartmentalisation in TAMR cells. They failed to even discuss potential mechanisms. Simply stating that it is "true" is not a mechanism. On a similar vein the authors were asked to determine or at least explain whether re-compartmentalization alters CTCF or vice versa. Their response that this is an interesting topic is inadequate.

The responses to many of the questions from the second reviewer also are not definitive and lack clarity.

The questions posed by the reviewers are critical to demonstrating the relevance, utility, robustness and generalizability of the data. They are further important for providing a mechanistic understanding of the data set. The authors need to review these and answer with clarity and rigor.

Reviewer #2 (Remarks to the Author):

While the authors made some small changes to the document, they did not address some of my major concerns. There are several concerns that remain unaddressed by the main two are below:

1) The authors still do not provide any estimate of False Discover Rate of differential compartmentalization nor control for it in any manner. This is critical.

2) The methods are still far too short and incomplete. The authors said that they updated the method section with 'much more details'. But as far as I can tell they only added 5 sentences. In their response to reviewers they explained how they did some things. For example, they explained how they performed differential ChIP-seq analysis with DESeq2. But they do not include that in the methods section. Incidentally, there is also no citation for DESeq2. This is just one small example. But the authors did a lot of work in this paper and the methods section is only 3.5 pages long. The authors should explain what they did in MUCH greater detail. While I am obviously more concerned with the content than the exact length, at this font size and spacing I would expect at least 10-15 pages of methods. It is not clear how the authors analyzed their data. And there is no way someone could reproduce these analyses based on these descriptions.

My main concern reviewing this paper (and any paper) is that the experiments are performed correctly with sound (doesn't have to be perfect) statistical analysis and that the methods are clearly described. The experiments themselves seem fine. But the statistical analysis and description thereof are lacking. Both fall far short of the bar required for publication.

Response to Reviewers

Reviewer #1 (Remarks to the Author):

1. *The abstract is modestly improved but still requires editing for English syntax and clarity.*

Response: We have significantly improved the abstract and made it much clear.

2. *The authors were asked to demonstrate generalizability of the data beyond the MCF7 cell line. Providing data in the rebuttal and not including this in the manuscript is not acceptable in today's climate where generalizability is critical. Indeed, there are marked differences between the results in T47D and MCF7 that are important to characterize and understand and importantly to determine which represent a generalizable event. Thus indicating that this information is beyond the scope of this work.*

Response: We now included the TCC data for both MCF7 and T47D cell lines. The new T47D TCC data were also deposited in GEO under accession number GSE119890. Interestingly, we observed the same trends of E2-induced time-dependent compartments in both MCF7 and T47D cell lines (Suppl. Figure S30), where the major trend is miscellaneous (dynamic changed) compartments in both MCF7 and T47D cell lines. Moreover, the altered compartments of both MCF7-TamR and T47D-TamR have higher percentage of miscellaneous compartments (Suppl. Figures S31, S32). Our results suggest that our analytical strategy and observations are generalizable in various cell lines. We have added it to the Discussion (Page 11).

3. *Both reviewers asked the authors to describe and develop clear and statistically robust methods to explain how the 24 patterns were defined and then how they were compartmentalized into six types of biological meaningful compartments. The response of the authors did not answer the question. How were the patterns defined and how were they compartmentalized and further how what "biologically meaningful" defined.*

Response: As described in Methods, Suppl. Methods and Fig. 2A, Suppl. Figure S14, S15, we defined 24 patterns or six types of dynamic changed compartments based upon how the re-compartmentalization in MCF7 cells respond to E2 induction. Highly common compartments (Pattern 1-4) are compartments that were neither changed in 1 hour E2-induced (T1) or in 24 hours E2-induced (T24) cells. Early transit compartments (Pattern 5-8) are ones that were changed in 1hr E2-induced but not in 24hr E2-induced cells. Late transit compartments (Pattern 9-12) are ones that were changed in 24hr E2-induced but not in 1hr E2-induced cells. And dynamic changed compartments (Pattern 13-24) are ones that were at least changed both in 1hr and 24hr E2-induced cells. In turn, this is a biological meaningful analytical approach. Interestingly, the resulted six types are identical to the analysis based on a mathematical calculation resulting in a total of 256 combinations of E2-induced time-dependent compartments

(Suppl. Figure S29). As demonstrated in Suppl. Figure S29, when using no E2 treatment (T0) as a contrast, there are four combinations for compartment changes from no E2 treatment to 1hr E2-induced: A→A, B→B, A→B, and B→A. Same can be obtained for 4hr, 16hr and 24hr E2-induced cells respectively. Thus, the total combinations of E2-induced time-dependent compartments are $4 \times 4 \times 4 \times 4 = 256$. For example, Pattern 1 can be interpreted as the combinations of no changes of compartments (A→A or B→B) in 1hr, 4hr, 16hr and 24hr E2-induced cells. Pattern 2 can be interpreted as the combinations of no changes of compartments (A→A or B→B) in 1hr, 16hr and 24hr E2-induced cells, but minor changes (A→B or B→A) in 4hr E2-induced cells. Pattern 16~24 can be interpreted as the combinations of changes of compartments (A→B or B→A) in all time points E2-induced cells.

4. The authors were asked to provide a reason why there would be greater re-compartmentalisation in TAMR cells. They failed to even discuss potential mechanisms. Simply stating that it is "true" is not a mechanism. On a similar vein the authors were asked to determine or at least explain whether re-compartmentalization alters CTCF or vice versa. Their response that this is an interesting topic is inadequate.

Response: There are seven possible mechanisms of tamoxifen resistance summarized by Chang et al. (2012): (1) Loss of ER expression and function; (2) Altered expression patterns of coregulatory proteins; (3) Growth factor receptors/kinase signal transduction pathways; (4) Pharmacological and metabolic aspects; (5) Regulation of autophagy and/or apoptosis; (6) ER-negative cancer stem cells; (7) Antioxidant protein-mediated cell survival, in which tamoxifen prevents repression of antioxidant proteins. Among them, No. 1-3 could be relevant to re-compartmentalization. Our observations suggest a possible molecular mechanistic model (Figure 7A-D) (Please refer to discussion section, Page 11-12). The constitutive estrogen stimulation in breast cancer cells enhances stronger ER α activity and further recruits its distal regulatory machinery including different co-regulators, mediators, cohesions and chromatin remodelers, and then mobilizes highly dynamic gene looping events which essentially expand to render a 3D genome re-compartmentalization meanwhile force CTCF eviction resulting in reduced insulation activity at the compartment boundary. In contrast, in breast cancer cells with the acquired resistance, increased crosstalk between ER α and other signal transduction pathways such as EGFR/HER2 (Massarweh et al., 2008), IGF-IR (Browne et al., 2013), and AKT/PTEN (Shoman et al., 2005) or altered expression of some key co-regulators particularly reshuffle these highly dynamic gene looping resulting in altered chromatin reorganization. In general, it is well known that CTCF plays an important role in mediating the 3D genome architecture in any cell lines. However, our data also supported a notion that ER α regulated highly re-compartmentalization may require to loosen CTCF insulator at the domain boundary which may be responsible for tamoxifen resistance in breast cancer cell lines. These are not contradictory because the relationship of CTCF and chromatin structures varies in different cell status.

References:

Minsun Chang. (2012) Tamoxifen Resistance in Breast Cancer. *Biomol Ther (Seoul)*. 2012 May; 20(3): 256–267.

Massarweh, S., Osborne, C.K., Creighton, C.J., Qin, L., Tsimelzon, A., Huang, S., Weiss, H., Rimawi, M., Schiff, R. (2008) Tamoxifen resistance in breast tumors is driven by growth factor receptor signaling with repression of classic estrogen receptor genomic function. *Cancer Res* 68, 826-833.

Browne, B.C., Hochgräfe, F., Wu, J., Millar, E.K., Barraclough, J., Stone, A., McCloy, R.A., Lee, C.S., Roberts, C., Ali, N.A., et al. (2013) Global characterization of signalling networks associated with tamoxifen resistance in breast cancer. *FEBS J*. 280, 5237-57.

Shoman, N., Klassen, S., McFadden, A., Bickis, M.G., Torlakovic, E. and Chibbar, R. (2005) Reduced PTEN expression predicts relapse in patients with breast carcinoma treated by tamoxifen. *Mod Pathol* 18, 250–259.

Reviewer #2 (Remarks to the Author):

1) *The authors still do not provide any estimate of False Discover Rate of differential compartmentalization nor control for it in any manner. This is critical.*

Response: We now provided an estimate of False Discover Rate (FDR) of the differential compartment between any two compartments by a permutation-based test. In brief, the difference in Means of Eigenvector Values of each of two compartments: Compartment 1 and Compartment 2, was first calculated as the Observed Value. The Eigenvector Values of two compartments were then pooled together and randomly selected one half as randomized Compartment 1 and the other half as randomized Compartment 2. This was done for a total of 1,000 rounds of permutation. In each round, the difference in means of Eigenvector Values of each of two randomized compartments was calculated as a Permuted Value. All Permuted Values were combined into a null distribution. The FDR is estimated based on how many Permuted Values are above the Observed Value and the permuted null. As such, FDRs of each type of compartments are the following: HCC 0.268; ETC 0.230; LTC 0.192; LDC 0.178; MDC 0.161; HDC 0.201; TA-HCC 0.154; TA-ETC 0.250; TA-LTC 0.154; TA-LDC 0.165; TA-MDC 0.139; TA-HDC 0.226. Both the method (Page 24) and results (Pages 6 and 7) were added into the text. These FDRs are very reasonable given that our initial analysis strategy was based upon how the re-compartmentalization in MCF7 cells respond to E2 induction, therefore we keep all of them for the downstream analysis.

2) *The methods are still far too short and incomplete. The authors said that they updated the method section with much more details. But as far as I can tell they only added 5 sentences. In their response to reviewers they explained how they did some things. For example, they*

explained how they performed differential ChIP-seq analysis with DESeq2. But they do not include that in the methods section. Incidentally, there is also no citation for DESeq2. This is just one small example. But the authors did a lot of work in this paper and the methods section is only 3.5 pages long. The authors should explain what they did in MUCH greater detail. While I am obviously more concerned with the content than the exact length, at this font size and spacing I would expect at least 10-15 pages of methods. It is not clear how the authors analyzed their data. And there is no way someone could reproduce these analyses based on these descriptions.

Response: We performed the differential ChIP-seq analysis with DESeq2 in Methods section of “Differential binding analysis of ER α ”. Now we further extended the Methods section in main text to five pages and added around 10 pages of Supplementary Methods section in Supplementary Material, so the total number of pages of Methods has been increased from 3.5 pages to 15 pages. In addition, there are brief descriptions of analyses included in some of Supplementary Figures.

Reviewers' comments:

Reviewer #2 (Remarks to the Author):

The authors have addressed my major concerns. They now use permutation testing to provide FDR control of their differential compartment analyses and they have now adequately described the methods used. I still find the paper a bit hard to interpret. There are just too many groups being compared and the meaning of the findings is not entirely clear to me. That being said, my main concern is whether or not the science was conducted properly and adequately described. I now think the authors did address those primary concerns.